# Repurposing type I–F CRISPR–Cas system as a transcriptional activation tool in human cells

Yuxi Chen[1,2,6], Jiaqi Liu[1,6], Shengyao Zhi[1], Qi Zheng[1], Wenbin Ma [1], Junjiu Huang[1,3], Yizhi Liu[2], Dan Liu[4], Puping Liang [1✉] & Zhou Songyang[1,2,4,5✉]

Class 2 CRISPR–Cas proteins have been widely developed as genome editing and transcriptional regulating tools. Class 1 type I CRISPR–Cas constitutes ~60% of all the CRISPR–Cas systems. However, only type I–B and I–E systems have been used to control mammalian gene expression and for genome editing. Here we demonstrate the feasibility of using type I–F system to regulate human gene expression. By fusing transcription activation domain to *Pseudomonas aeruginosa* type I–F Cas proteins, we activate gene transcription in human cells. In most cases, type I–F system is more efficient than other CRISPR-based systems. Transcription activation is enhanced by elongating the crRNA. In addition, we achieve multiplexed gene activation with a crRNA array. Furthermore, type I–F system activates target genes specifically without off-target transcription activation. These data demonstrate the robustness and programmability of type I–F CRISPR–Cas in human cells.

[1] The Second Affiliated Hospital, Sun Yat-sen University; MOE Key Laboratory of Gene Function and Regulation and Guangzhou Key Laboratory of Healthy Aging Research, School of Life Sciences, Sun Yat-sen University, Guangzhou 510275, China. [2] State Key Laboratory of Ophthalmology, Zhongshan Ophthalmic Center, Sun Yat-sen University, Guangzhou 510060, China. [3] Key Laboratory of Reproductive Medicine of Guangdong Province, the First Affiliated Hospital and School of Life Sciences, Sun Yat-sen University, Guangzhou 510275, China. [4] Verna and Marrs Mclean Department of Biochemistry and Molecular Biology, Baylor College of Medicine, One Baylor Plaza, Houston, TX 77030, USA. [5] Guangzhou Regenerative Medicine and Health-Guangdong Laboratory (GRMH-GDL), Guangzhou 510530, China. [6]These authors contributed equally: Yuxi Chen, Jiaqi Liu. ✉email: liangpp5@mail.sysu.edu.cn; songyanz@mail.sysu.edu.cn

Clustered regularly interspaced short palindromic repeats (CRISPR) and CRISPR-associated (cas) genes-based defence systems protect bacteria and archaea against phage and other foreign genetic elements[1–3]. Since the identification of increasing number of cas genes, the CRISPR–Cas systems have been classified into two Classes (Class 1 and Class 2) and six types (Type I–VI)[4] based on the different arrangements of cas genes and the subunits of effector complexes[5–7]. Class 2 CRISPR–Cas systems, the best-studied system with single effector protein (e.g., Cas9, Cas12, or Cas13) for foreign DNA or RNA interference, are subdivided into Type II (Cas9), Type V (Cas12), and Type VI (Cas13). In the past few years, Class 2 CRISPR–Cas systems have revolutionized both basic and clinical researches, enabling more rapid, precise, and robust genome editing and modifications in cultured cells and animals[8–17]. However, there were only a few applications of Class 1 CRISPR–Cas (Type I, Type III and Type IV) system.

Class 1 type I CRISPR–Cas systems are the most prevalent (~60%) in both bacteria and archaea, whereas class 2 only makes up ~10% of all CRISPR–Cas systems[18,19]. Differing from the Class 2 CRISPR–Cas systems, the Class 1 type I system relies on Cascade (CRISPR-associated complex for antiviral defense complex) for DNA binding, which further recruits Cas3 to degrade the foreign DNA[20]. Cascade, which recognizes and binds specific DNA, is a complex consist of multiple Cas proteins and CRISPR RNA (crRNA). CRISPR–Cas expression involves cas genes expression and CRISPR transcription, yielding a precursor crRNA (pre-crRNA). The pre-crRNA is processed at the repeat regions by Cse3[3], Cas6[21] or Csy4[22] to generate mature crRNA with different characteristics. Other Cas proteins then bind onto the crRNA and assemble into a functional Cascade[23–26]. Cascade discriminates the self and non-self DNAs by recognizing the PAM (proto-spacer adjacent motif) sequence[27], which triggers a conformational change upon binding[28,29]. The conformational change finally recruits Cas3 for invasive DNA degradation[20,30–32].

Compared to the widely used class 2 CRISPR–Cas systems, the multiple-subunit class 1 type I CRISPR–Cas system has distinct properties, for example, generating large fragment deletion in genome editing with Cas3[33,34], and multiple subunits for different Cas protein–effector fusion strategies[35]. These differences between the class 1 and class 2 CRISPR–Cas system may contribute to the advantages of Class 1 CRISPR–Cas system in some applications. Accroding to recent classification studies, there are seven subtypes (I–A to I–G) in type I CRISPR–Cas system[7,36]. In recent years, the type I–A[37], I–B[38,39], I–E[40], and I–F[41,42] CRISPR–Cas have been used for prokaryotic gene engineering in Sulfolobus islandicus (I–A), Clostridium pasteurianum (I–B), Lactobacillus crispatus (I–E), Zymomonas mobilis (I–F), and Pseudomonas aeruginosa (I–F). Besides, type I–B[43] and type I–E[44–46] Cascades can work as transcription repressor in Sulfolobus islandicus (I–B) and Escherichia coli (I–E). Furthermore, type I–E and I–B CRISPR–Cas systems have been used in human cells[33–35,47] and plants[48] for gene editing and transcription regulation. Therefore, developing tools based on type I CRISPR–Cas system might provide alternative tools for genome editing and gene regulation.

Type I–F CRISPR–Cas system is among the well-studied CRISPR–Cas systems. It has fewer Cascade components than type I–E CRISPR–Cas system (4 vs 5), which will be easier to be controlled and delivered. The type I–F CRISPR–Cas system was first discovered as CRISPR subtype Ypest from Yersinia pestis[49,50]. The Cascade components of type I–F CRISPR–Cas system were also named as Csy (CRISPR subtype Ypest) subunits, which includes Csy1 (Cas8f1), Csy2 (Cas5f1), Csy3 (Cas7f1), and Csy4 (Cas6f)[7,26] (Fig. 1a). In addition, the Cascade of type I–F

variant (type I–Fv, or type I–F2) CRISPR–Cas system, derived from type I–F system, consists of only three subunits: Cas5fv (Cas5f2), Cas6f, and Cas7fv (Cas7f2)[4,7] (Fig. 1a). The type I–F and type I–Fv Cascade recognizes 5′-CC PAM on the non-target strand for target binding[51,52]. Their crRNAs consist of 8-nt 5′ handle for Csy1 and Csy2 binding, 32-nt spacers bound by six copies of Csy3 for target recognition, and 20-nt 3′ hairpin for Csy4 binding and pre-crRNA processing[22]. Recently, type I–F CRISPR–Cas system has been used for genome engineering in Zymomonas mobilis[41] and Pseudomonas aeruginosa[42]. However, there has not been any report on the exploitation of the type I–F or type I–Fv CRISPR–Cas system for genome manipulation application in human cells yet.

In this study, we explore the possibility of developing programmable type I–F and type I–Fv CRISPR tools for transcription activation in mammalian cells. In contrast to type I–E and I–B, Pseudomonas aeruginosa type I–F and Shewanella putrefaciens type I–Fv systems require fewer subunits for dsDNA targeting in bacteria[53,54]. Also, the multiple subunits in type I–F and type I–Fv might provide different combinations for tagging and increase signal strength when genetic modulators are fused to different subunits. By fusing the VPR (VP64-p65-Rta) transcription activation domain to the type I–F Cascade subunit Csy3, we achieve both exogenous (e.g., GFP expression) and endogenous (e.g., HBB, HBG1/2, SOX2, OCT4, IL1B, and IL1R2) gene activation in HEK293T cells. Interestingly, by changing the spacer length of crRNA, we can enhance the activation level of target genes. As is the case for class 2 systems, we can achieve multiplex gene activation through a customized CRISPR array from a single vector. Finally, the type I–F CRISPR–Cas system can activate target genes specifically without altering the expression of any predicted off-target genes. These data demonstrate the feasibility of using type I–F CRISPR–Cas system for programmable transcription activation and may have important implications in their adaptation for genome editing.

## Results

**Type I–F CRISPR–Cas maintains activity in human cells.** Csy1, Csy2, Csy3, and Csy4 constitute the Cascade complex in the Pseudomonas aeruginosa type I–F CRISPR–Cas system (PaeCascade) (Fig. 1a)[26,53,55]. Csy1 mediates PAM recognition (5′-CC-3′) at the 5′ end of the protospacer. Csy1 and Csy2 bind to the 5′ handle of the crRNA. Multiple Csy3 binds to the crRNA, serving as the backbone of the complex (Fig. 1b). Each Csy3 binds to 6-nt of the crRNA spacer with the precise number of Csy3 subunits determined by the length of the crRNA spacer[56] (from 14 to 50-nt), resulting in 3–9 copies of Csy3[57]. Csy4 binds to the crRNA 3′ hairpin structure and is responsible for pre-crRNA maturation (Fig. 1b). In comparison, Shewanella putrefaciens type I–F variant Cascade (SpuCascade) contains only three subunits (Cas5fv, Cas6f, and Cas7fv) (Fig. 1a), leading to its more open configuration (Supplementary Fig. 1)[54]. Here, Cas5fv plays an important role in PAM recognition and dsDNA unwinding. Casf7v is involved in crRNA-target ssDNA duplex and non-target ssDNA binding to stabilize the complex, while Cas6f participates in pre-crRNA processing and crRNA hairpin binding (Fig. 1c).

We first expressed and purified PaeCascade and SpuCascade complexes in E. coli to test their dsDNA binding ability by electrophoretic mobility shift assays (EMSA). As shown in Fig. 1b, c, both PaeCascade and SpuCascade complexes could shift the dsDNA target probe (crRNA) in vitro. Next, we examined the expression of individual PaeCascade and SpuCascade subunits in 293T cells (Supplementary Fig. 2). While the level of expression differed between subunits, they could all be readily expressed in

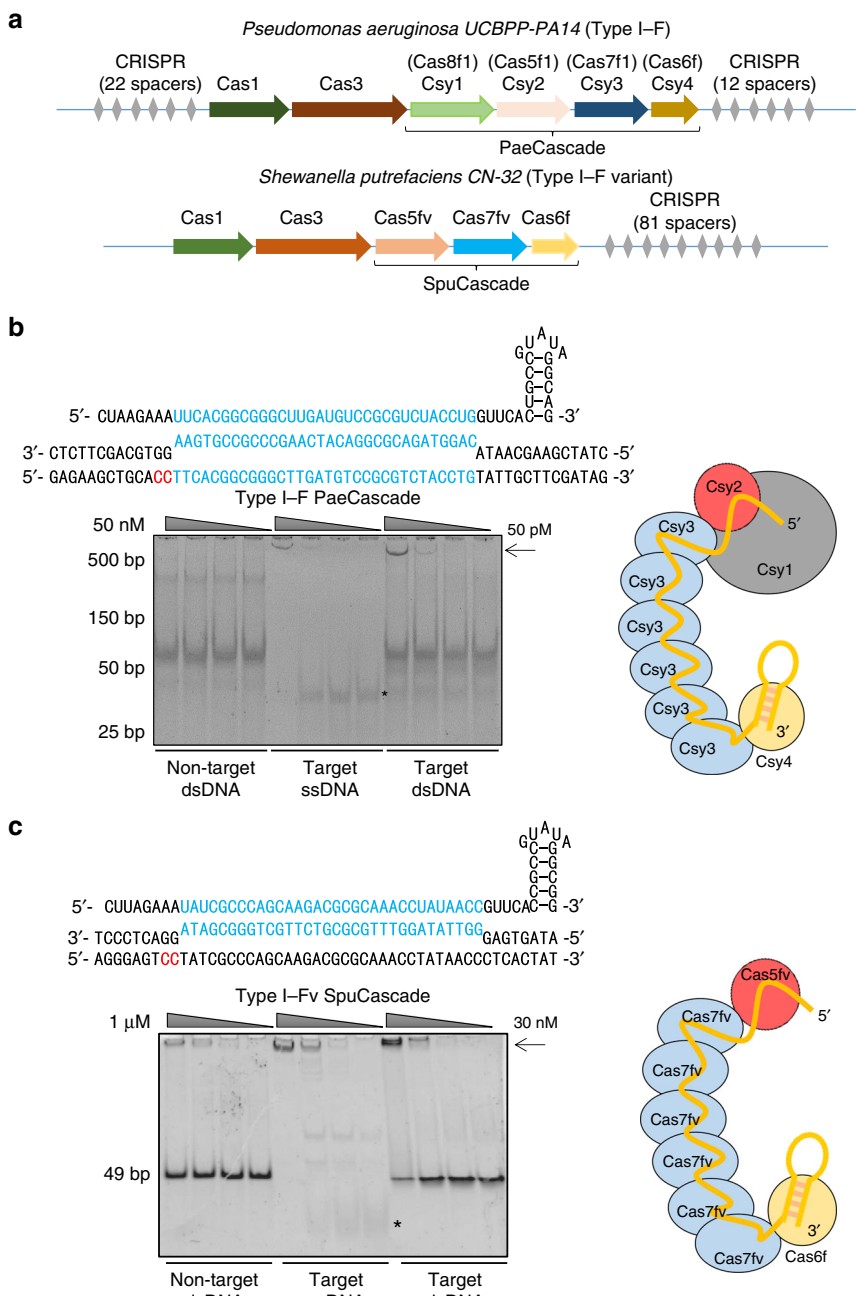

**Fig. 1** (continued)

mammalian cells. Both Csy4 and Cas6f are involved in crRNA maturation by processing the direct repeat (DR) of pre-crRNA[55,57]. We, therefore, tested the activities of ectopically expressed Csy4 and Cas6f using HEK293T cells transiently expressing a DR-GFP fusion sequence (DR-GFP) (Supplementary Fig. 3a). When DR-GFP was co-expressed with Csy4 or Cas6f, the percentages of GFP positive cells were drastically reduced (Supplementary Fig. 3b), indicating successful cleavage of the DR-GFP fusion mRNA.

**Targeted transcription activation by type I–F CRISPR–Cas.** To better examine PaeCascade and SpuCascade, we introduced rtTA (reverse tetracycline-controlled transactivator) expression cassette and eGFP expression cassette controlled by a minimal CMV promoter plus six copies of the tetracycline-responsive element

(TRE) into HEK293T cells by lentiviral vector (TRE-eGFP reporter) (Fig. 1d). When dCas9-VPR (dCas9 fused to transcription activator VP64-p65-Rta[58]) was co-transfected with gRNAs targeting the TRE sequence into TRE-eGFP reporter cells, percentages of GFP positive cells were significantly increased, indicating successful targeting of dCas9-VPR to the promoter and transcriptional activation of eGFP (Supplementary Fig. 4). With the TRE-eGFP reporter cells, we wanted to test whether PaeCascade and SpuCascade can bind dsDNA and induce transcription activation in mammalian cells. We next fused VPR to each of the codon-optimized PaeCascade and SpuCascade subunits and generated polycistronic all-in-one expression vectors of the Cascade complexes. To test possible effects due to configuration differences, we generated vectors with the same subunits in different sequences (Fig. 1e, f and Supplementary Fig. 5). Then we tested their activity in the TRE-eGFP reporter cells together

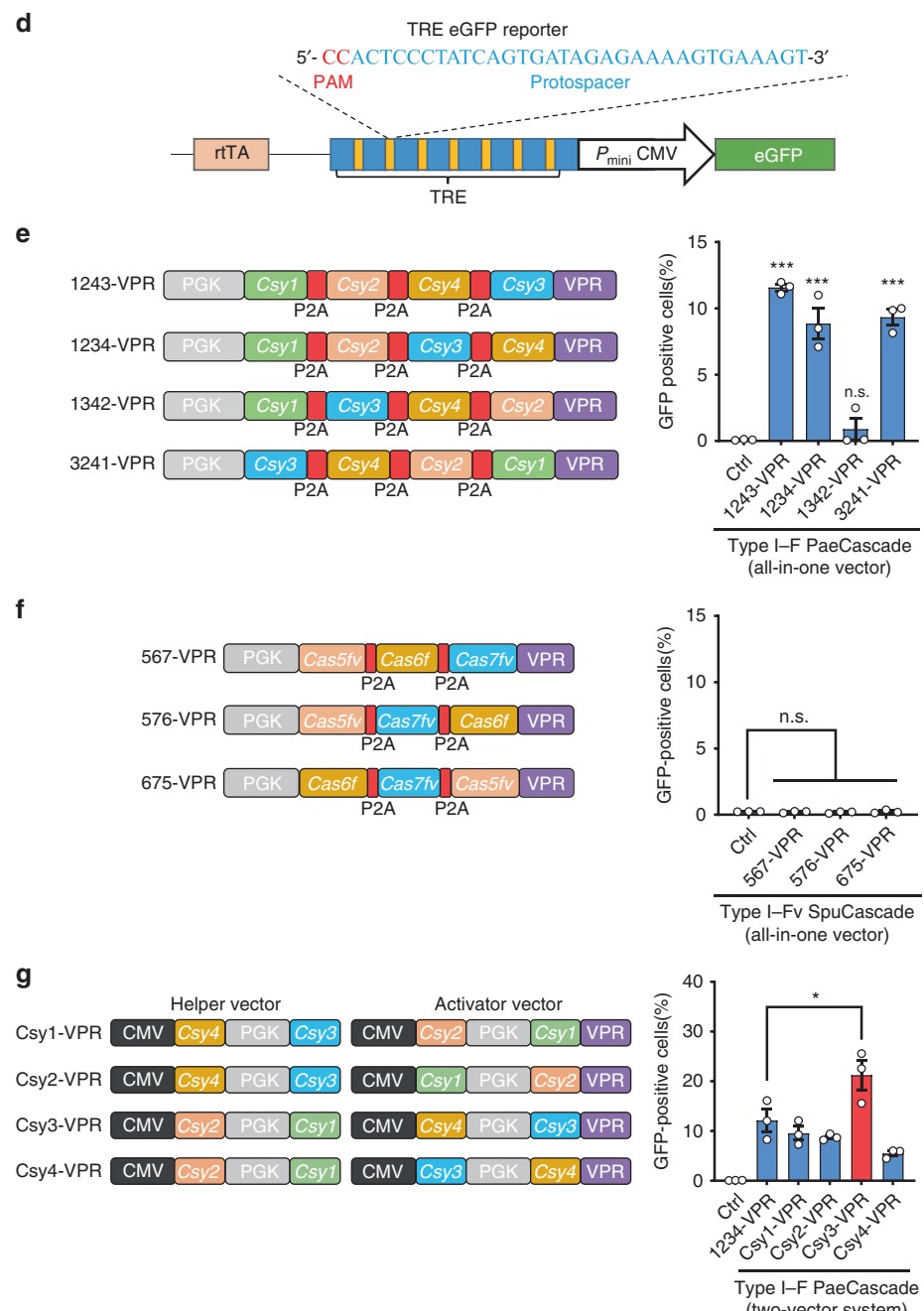

**Fig. 1 Targeted transcription activation by type I–F PaeCascade. a** Schematic diagram of the *Pseudomonas aeruginosa* type I–F and *Shewanella putrefaciens* type I–Fv CRISPR–Cas locus. Cas proteins are presented with arrows in different colors. CRISPR repeats are indicated with gray diamonds. **b** Electrophoresis mobility shift assays to detect target DNA binding by PaeCascade. Up, schematic representation of the processed crRNA with 5′-CC-3′ PAM recognition and base pairing at the DNA target site. Down, the result of the EMSA assay. The arrow indicates PaeCascade–crRNA–DNA complex. "*" Indicates free ssDNA. **c** Electrophoresis mobility shift assays to detect target DNA binding by SpuCascade. Up, schematic representation of the processed crRNA with 5′-CC-3′ PAM recognition and base pairing at the DNA target site. Down, the result of the EMSA assay. The arrow indicates the SpuCascade–crRNA–DNA complex. "*" Indicates free ssDNA. **d** A schematic of the integrated sequence in the TRE-eGFP reporter cell. The target sequences of type I–F and type I–Fv CRISPR–Cas system containing a 5′-CC-3′ PAM is shown. PAM is in red, and the target sequence is in blue. rtTA: reverse tetracyclin-transactivator. TRE: tetracyclin response element. **e** Flow cytometric analysis of GFP activation in TRE-eGFP reporter cells transfected with type I–F PaeCascade all-in-one vectors and crRNA expression vectors. Left: the all-in-one constructs used in the experiment. PaeCascade subunits linked by self-cleaving P2A peptides was driven by PGK promoter. Right: Quantification of GFP positive cell induced by type I–F PaeCascade all-in-one vectors. **f** Flow cytometric analysis of GFP activation in TRE-eGFP reporter cells transfected with type I–F SpuCascade all-in-one vectors and crRNA expression vectors. **g** Flow cytometric analysis of GFP activation in TRE-eGFP reporter cells transfected with type I–F PaeCascade 2-vector systems and crRNA expression vectors. Left: 2-vector systems used in the experiments. Right: quantification of GFP positive cells induced by type I–F PaeCascade 2-vector systems. Ctrl: untransfected control. Data represented three biological repeats and displayed as mean ± S.E.M. Statistical significance was calculated using one-way ANOVA (n.s., not significant; *$P < 0.05$; ***$P < 0.001$). Source data are provided as a Source Data file.

with a TRE-targeting crRNA. Three configurations of the ectopically expressed PaeCascade complex (1243-VPR, 1234-VPR, and 3241-VPR) were able to activate GFP expression in ~10% of the cells (Fig. 1e). In contrast, despite having fewer subunits, none of the SpuCascade vectors could activate GFP expression (Fig. 1f). Such differences reaffirm the notion that Cascade complexes have distinct properties from one another and warrant further mechanistic studies. Given the complicate chromatin structure of eukaryote into consideration (e.g., histone binding, different histone modification, and etc.), such distinct properties may due to their difference of PAM recognition mechanism (e.g., DNA minor groove vs major groove) and DNA helicase activity[54]. In the following sections, we will focus on type I–F PaeCascade and investigate how to use it to effectively and efficiently activate transcription.

Given the presence of multiple copies of Csy3 in a functional Cascade complex, Csy3 may become limiting during complex assembly if all subunits are encoded by a single transcript (Fig. 1e). To address this possibility, we devised a helper-activator strategy (Fig. 1g). Here, two helper vectors encode the subunits in pairs (Csy1/2 or Csy3/4). Activator vectors encode the subunits in pairs and have one of the subunits fused to VPR, resulting in four different activators in all (Supplementary Fig. 6). The activator vectors were then co-transfected into the TRE-eGFP reporter cells in combination with a helper vector and TRE-targeting crRNA (Fig. 1f). Among all the fusion types in the helper-activator 2-vector system, Csy1-VPR, Csy2-VPR, Csy3-VPR, and Csy4-VPR, only the Csy3-VPR fusion 2-vector system have a higher activating efficiency than the all-in-one 1234-VPR vector system. Further experiments with another two plasmid system, in which Csy1, Csy2, and Csy4 were expressed by P2A fusion in one plasmid and Csy3-VPR in another, showed highest HBB and HBG activation level in molar ratio = 1:3 (Supplementary Fig. 7a). However, its activation efficiency in HBG was not as good as Csy3-VPR fusion helper-activator 2-vector system (Supplementary Fig. 7b). Therefore, we decided to use the Csy3-VPR fusion helper-activator 2-vector system for further studies. And in the hitherto described experiments, Csy3-VPR refers to the Csy3-VPR fusion helper-activator 2-vector system. These data clearly showed that type I–F PaeCascade could be utilized to activate reporter gene expression.

**Endogenous gene activation by type I–F CRISPR–Cas.** Unlike most endogenous genes, multiple copies of TREs targeted by Cascade/crRNA existed in the TRE-eGFP reporter cells. To investigate PaeCascade-mediated transcriptional activation of endogenous genes, we designed a crRNA against ~200 bp upstream of the transcriptional start site (TSS) of the hemoglobin β protein coding gene (HBB). We co-transfected the crRNA expressing vector into HEK293T cells with the Csy3-VPR helper-activator vectors described above. Again, cells expressing the combination with Csy3-VPR fusion showed the highest transcription activation activity at the HBB locus (~15 fold higher than control cells) (Fig. 2a). For convenience, PaeCascade-VPR referred to Csy3-VPR in the test below. To determine how PaeCascade VPR fusion complex may differentially activate gene expression at different loci, we picked six genes (HBB, HBG, SOX2, OCT4, IL1B, and IL1R2) and designed crRNAs targeting different promoter regions (−500 bp to −100 bp upstream TSS) in each locus. In all cases, PaeCascade-VPR was able to activate endogenous gene expression to varying degrees (Fig. 2b), with the region 100–200 bp upstream of TSS being the best targets (Fig. 2c). And the fold activation of each gene was highly correlated to their basal expression level, with the weaker expressed genes showed greater fold change (Fig. 2d). These findings indicated

that the type I–F PaeCascade complex could robustly activate endogenous gene transcription.

Is PaeCascade-VPR more efficient than gene activation tools based on other CRISPR systems? To answer this question, we compared PaeCascade-VPR system to the other gene activation tools (dCas9-VPR, dAsCas12a-VPR, and type I–E EcoCascade-VPR). We designed crRNAs or gRNAs of these systems targeting to the same loci of HBB, HBG, SOX2, and IL1B (Fig. 3a). The results showed that dCas9-VPR had the highest transcription activity for HBB when targeting 170 bp upstream TSS (Fig. 3b). Except for the HBB -170bp TSS locus, PaeCascade-VPR appeared to outperform dCas9-VPR at activating transcription for gene loci examined (Fig. 3b–e). In all the loci tested, PaeCascade-VPR showed higher activating efficiency than dAsCas12a-VPR and EcoCascade-VPR (Fig. 3b–e). These data suggested that PaeCascade-VPR may be more efficient than canonical dCas9-VPR and other CRISPR-based systems, at least at certain gene loci, and represented a worthy addition to molecular tools that could modulate gene expression.

**Enhancing transcription activation through crRNA engineering.** Since the spacer length of PaeCascade crRNA may be extended (beyond the canonical 32-nt) to accommodate more Csy3 subunits (more Csy3-VPR)[53,54], we investigated the effect of spacer length on PaeCascade-VPR activity at the HBB, HBG, and SOX2 loci (Fig. 4a). Given that the minimal length for Cys3 binding is 6-nt, we varied the length of spacers by multiples of six. In each case, crRNAs with longer spacers (e.g., 50 and 56-nt) led to more efficient transcriptional activation (Fig. 4a), pointing to a simple yet effective way to regulate and tune endogenous gene expression through enriching VPR in a certain locus. To test whether Cascade-mediated transcriptional activation could be further manipulated, we co-transfected two crRNAs that target the same locus with the PaeCascade-VPR complex into cells. Among the six genes tested (HBB, HBG, SOX2, OCT4, IL1B, and IL1R2) (Distances between crRNAs: HBB crRNA1-crRNA2: 27 bp; HBG crRNA1-crRNA2: 55 bp; SOX2 crRNA1-crRNA2: 96 bp; OCT4 crRNA1-crRNA2: 79 bp; IL1B crRNA1-crRNA2: 71 bp; IL1R2 crRNA1-crRNA2: 65 bp), synergistic activation could be observed at four loci (Fig. 4b), indicating that simultaneous targeting of the PaeCascade-VPR complex to multiple regions of a promoter may enhance its activity. Not surprisingly, the distance between the two crRNA target regions also had an impact on the extent of transcriptional activation. We designed pairs of crRNAs with different distances and tested their ability to activate HBG expression (Fig. 4c). A distance about 50–75 bp appeared optimal for the HBG gene in this case. These observations underlined the multiple ways by which PaeCascade-VPR may be further improved as a robust and efficient tool for gene expression modulation.

**Multiplexed gene activation by customized CRISPR arrays.** The Pseudomonas aeruginosa CRISPR arrays, which contain tandem spacers linked by direct repeats (DRs), are transcribed and then processed by Csy4 to generate mature crRNAs that can target different sites[22]. We, therefore, reasoned that using customized CRISPR arrays driven by a single Pol. III promoter (e.g., hU6) might allow PaeCascade-VPR to bind multiple regulatory sites and achieve more efficient single gene activation. To this end, we constructed a vector that should yield a single transcript with spacer 1 and 2 that was subsequently processed by Csy4 into two mature crRNAs (Fig. 5a). Then, we constructed the CRISPR array expressing vectors to produce two crRNAs that target the gene loci of HBB, HBG, and SOX2 in HEK293T (Fig. 5b). In each case, introducing a single construct containing the CRISPR array could provide a transcriptional activation level comparable to that using

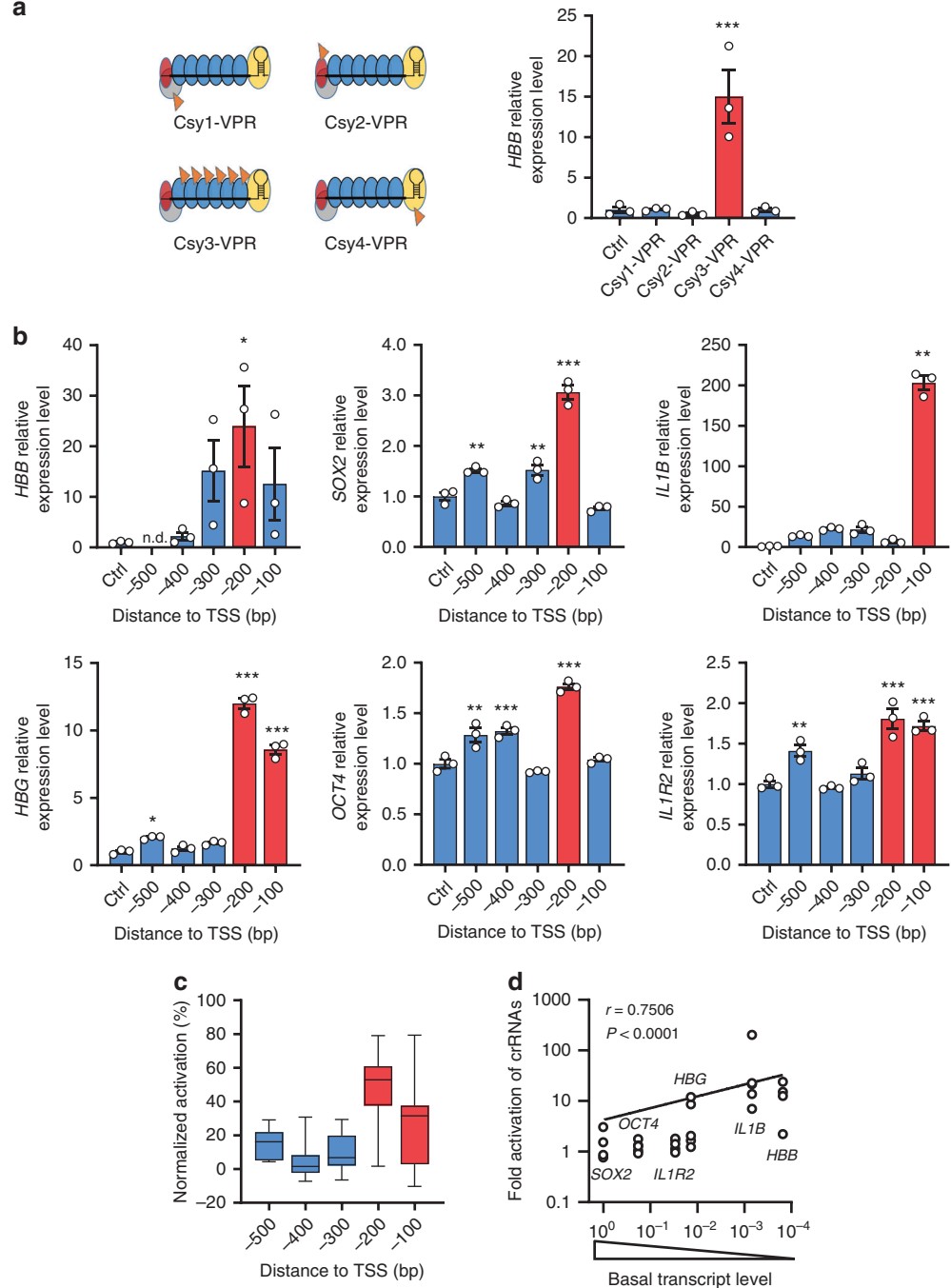

**Fig. 2 Type I–F PaeCascade could activate endogenous genes in human cells. a** Quantitative PCR analysis of *HBB* transcription level in HEK293T cells transfected with type I–F PaeCascade 2-vector systems and crRNA targeting *HBB*. Left: schematic illustration of different type I–F PaeCascade VPR activators generated from 2-vector systems in Fig. 1f. Gray: Csy1; red: Csy2; blue: Csy3; yellow: Csy4; orange flag: VPR. Different fusions of PaeCascade subunits resulted in different locations and copy numbers of VPR. HEK293T cells were transfected with PaeCascade 2-vector systems and crRNA vector. 48 h post-transfection, cells were lysed for RNA extraction and quantitative PCR assay. **b** Quantitative PCR analysis of gene transcription levels in HEK293T cells transfected with type I–F PaeCascade VPR (Csy3-VPR 2-vector system) targeting different regions upstream the transcriptional start site (TSS) of six genes (*HBB*, *HBG*, SOX2, *OCT4*, *IL1B*, and *IL1R2*). n.d.: not determined. **c** Histogram showing the normalized mean transcription activating levels of *HBB*, *HBG*, SOX2, *OCT4*, *IL1B*, and *IL1R2* induced by type I–F PaeCascade VPR (Csy3-VPR 2-vector system) transcription activator targeting different regions upstream of TSS. For normalization of data from different TSS among different genes, data in the same gene were processed by percentage normalization with 100% defined by the sum of all values in data set. The normalized values of all six genes were pooled and plotted as box & whiskers plot with min to max option. The median value is displayed as the center of the data set, and is derived using the lower and upper quartile values. The maximum and minimum values are displayed as whiskers. **d** The efficiency of target gene activation as a function of basal transcript levels. Data from (**b**) were plotted by fold changes comparing to negative control and relative basal transcript level of *HBB*, *HBG*, SOX2, *OCT4*, *IL1B*, and *IL1R2*. Each dot represented the mean relative activation level of each crRNA from the three replications in (**b**). Ctrl: non-targeting crRNA control. Data in (**a**, **b**) represented three biological repeats and displayed as mean ± S.E.M. Statistical significance was calculated using one-way ANOVA (*$P < 0.05$; **$P < 0.01$; ***$P < 0.001$). Source data are provided as a Source Data file.

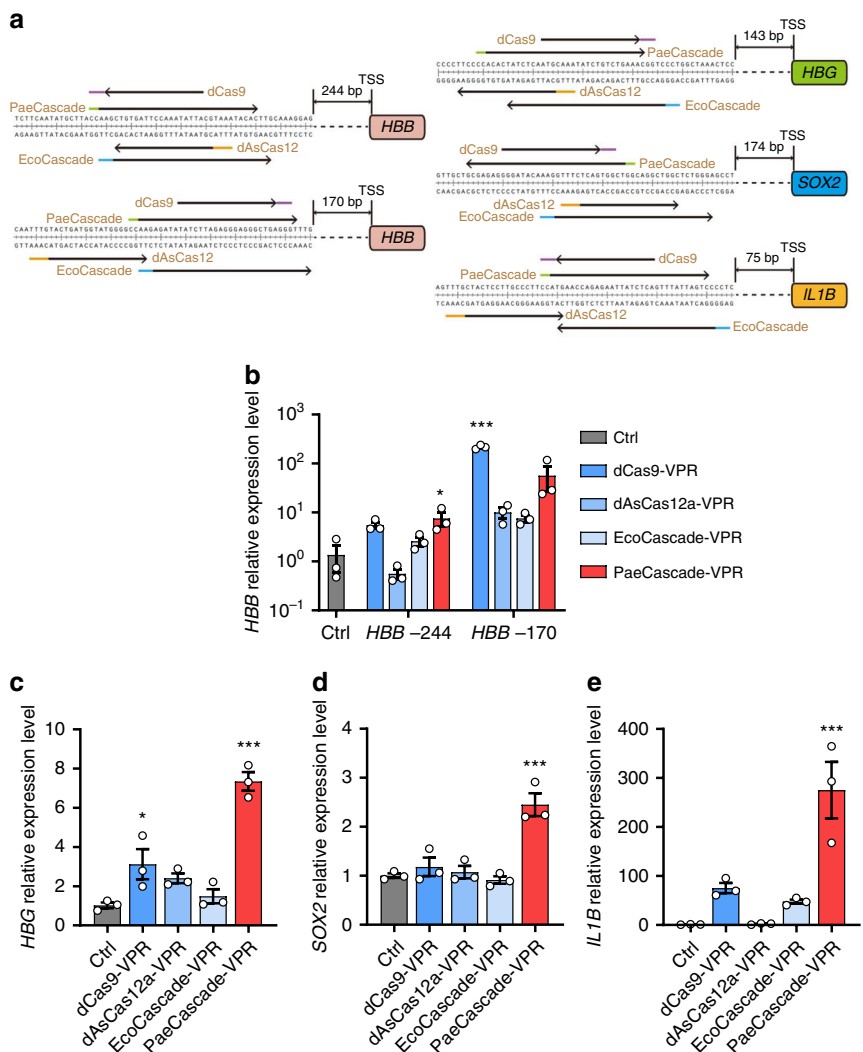

**Fig. 3 Comparison of type I–F PaeCascade and other VPR-based systems. a** Schematic illustrating of the gRNAs and crRNAs of dCas9, dAsCas12a, EcoCascade, and PaeCascade targeting different regions upstream the transcriptional start site (TSS) of four genes. Arrow directions indicated the 5′ to 3′ sequence of the spacer. Colored lines (purple, green, yellow, and blue) represented the PAM motif. **b** Quantitative PCR analysis of *HBB* transcription levels in HEK293T cells transfected with dCas9-VPR, dAsCas12a-VPR, EcoCascade-VPR, or PaeCascade VPR (Csy3-VPR). **c** Quantitative PCR analysis of *HBG* transcription levels in HEK293T cells transfected with dCas9-VPR, dAsCas12a-VPR, EcoCascade-VPR, or PaeCascade VPR (Csy3-VPR). **d** Quantitative PCR analysis of *SOX2* transcription levels in HEK293T cells transfected with dCas9-VPR, dAsCas12a-VPR, EcoCascade-VPR, or PaeCascade VPR (Csy3-VPR). **e** Quantitative PCR analysis of *IL1B* transcription levels in HEK293T cells transfected with dCas9-VPR, dAsCas12a-VPR, EcoCascade-VPR, or PaeCascade VPR (Csy3-VPR). Ctrl: non-targeting crRNA control. Data represented three biological repeats and displayed as mean ± S.E.M. Statistical significance was calculated using one-way ANOVA (*$P < 0.05$; ***$P < 0.001$). Source data are provided as a Source Data file.

two individual crRNA vectors (Fig. 5b). Furthermore, the same strategy could be used to produce spacers that target different genes (at least three genes) simultaneously and effectively activate gene transcription (Fig. 5c). The ability of PaeCascade-VPR to activate multiplex genes simultaneously with a customized CRISPR array in a single construct instead of individual crRNAs in independent constructs enormously simplified the activation system, which increased the transfection efficiency and makes it not necessary to express and deliver multiplex gRNAs independently in comparison with type II CRISPR system. These data pointed to PaeCascade-VPR as a powerful and flexible system with much untapped potential for research applications compared to the much better-studied type 2 systems.

**Mismatch and off-target analysis of PaeCascade-VPR system.** Although the DNA-binding property of PaeCascade is crucial for

its specificity in mammalian cells, it remains poorly understood. It has been shown that the seed region (first 8-nt of PAM proximal sequence) within the crRNA is critical for initiating target binding and DNA unwinding[26]. To further probe the target DNA binding specificity of PaeCascade in mammalian cells, we generated a series of *HBB* and *HBG* targeting crRNA variants with 6-nt mismatches in the 32-nt spacer region (Fig. 6a). Being consistent with previously published data from in vitro experiments[26], mismatches in PAM-proximal regions had the biggest impact on the activity of PaeCascade-VPR, with cells exhibiting the lowest activation levels of *HBB* and *HBG* with these crRNA variants (Fig. 6b). Next, we constructed 32 crRNA variants with single-nucleotide mismatches in the 32-nt spacer to determine the contribution of each position (Fig. 6c). As shown in Fig. 5d, mismatches at nearly every position reduced the level of gene activation. Again, changes in PAM-distal positions had less impact on *HBB*/*HBG* activation than those at PAM-proximal

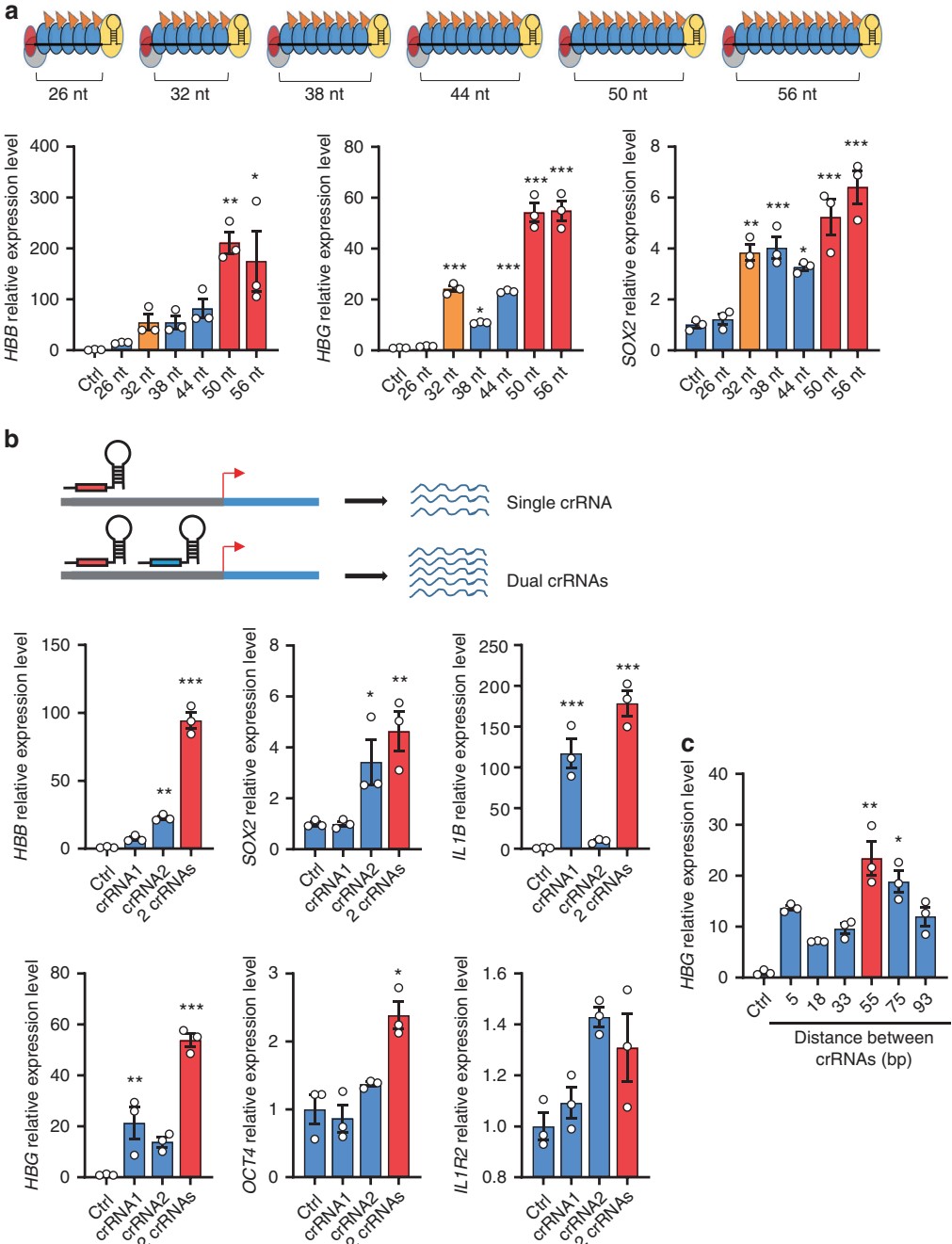

**Fig. 4 crRNA engineering to enhance the activation of type I–F PaeCascade. a** Quantitative PCR analysis of *HBB, HBG,* and *SOX2* transcription levels in HEK293T cells transfected with type I–F PaeCascade VPR (Csy3-VPR 2-vector system) with spacers in different lengths. Upper: schematic illustration of differences in Csy3 copy numbers in type I–F PaeCascade VPR (Csy3-VPR) with spacers in different lengths. Gray: Csy1; red: Csy2; blue: Csy3; yellow: Csy4; orange flag: VPR. The longer the spacer was, the more Csy3-VPR in type I–F PaeCascade. Lower: quantitative PCR analysis of *HBB, HBG,* and *SOX2* transcription level in HEK293T cells cotransfected with type I–F PaeCascade VPR and crRNA targeting *HBB, HBG,* and *SOX2*. 48 h post-transfection, cells were lysed for RNA extraction and quantitative PCR assay. **b** Quantitative PCR analysis of *HBB, HBG,* SOX2, OCT4, IL1B, and IL1R2 transcription levels in HEK293T cells transfected with type I–F PaeCascade VPR (Csy3-VPR 2-vector system) and crRNAs targeting −100 bp (crRNA1) and −200 bp (crRNA2) upstream of TSS in Fig. 2. Upper: schematic illustration of enhancing transcription level by two crRNAs targeting the same gene. Lower: quantitative PCR analysis of *HBB, HBG,* SOX2, OCT4, IL1B, and IL1R2 transcription levels in HEK293T cells. 2 crRNAs indicates two independent crRNA expression vectors. **c** Quantitative PCR analysis of *HBG* transcription level in HEK293T cells cotransfected with type I–F PaeCascade VPR (Csy3-VPR) and crRNA with different distances to crRNA2 (−200 bp upstream TSS in Fig. 2). 48 h post-transfection, cells were lysed for RNA extraction and quantitative PCR assay. Ctrl: non-targeting crRNA control. Data represented three biological repeats and displayed as mean ± S.E.M. Statistical significance was calculated using one-way ANOVA (\*$P < 0.05$; \*\*$P < 0.01$; \*\*\*$P < 0.001$). Source data are provided as a Source Data file.

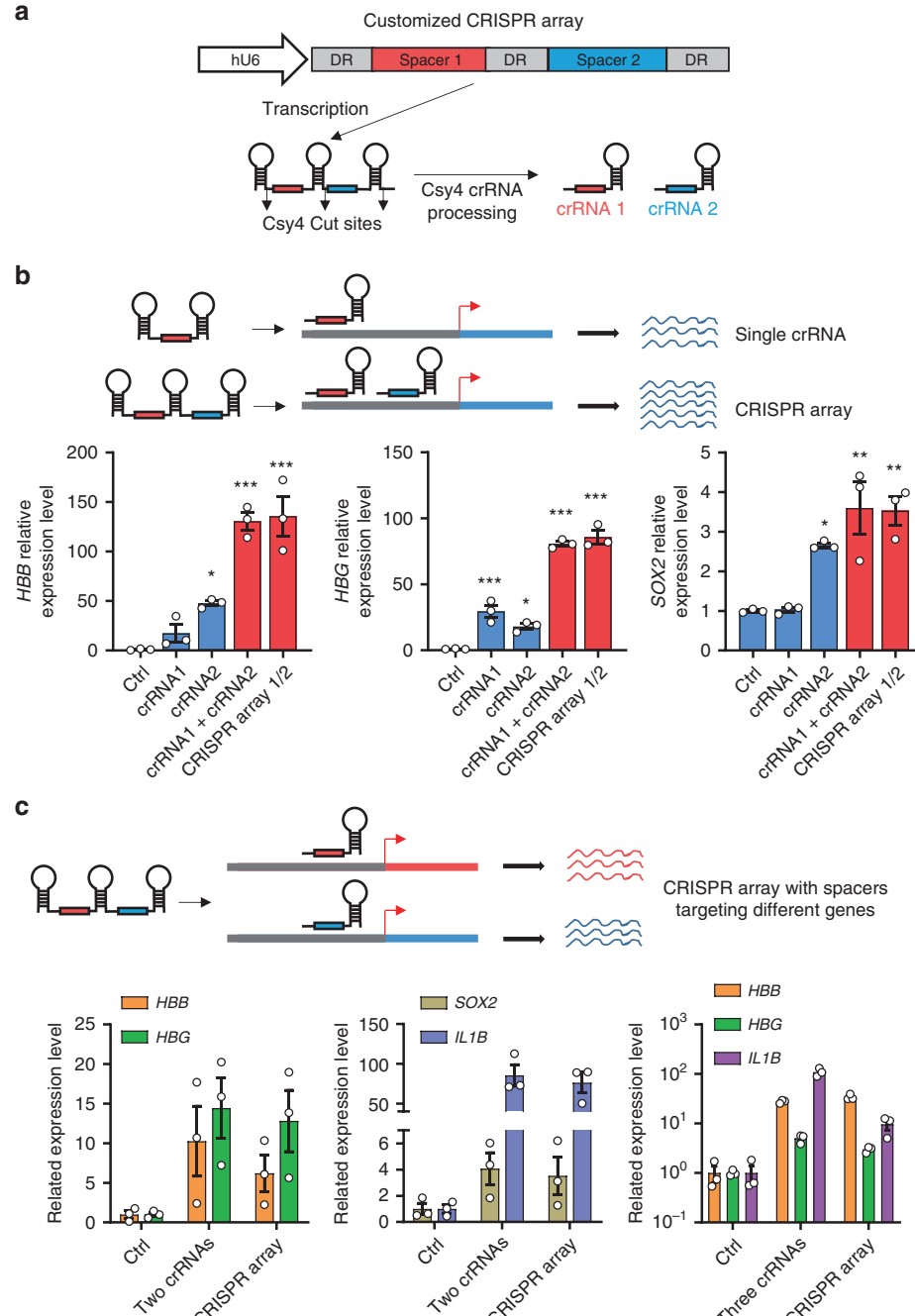

**Fig. 5 Transcription activation with customized CRISPR arrays by type I–F PaeCascade. a** Schematic illustrating pre-crRNA processed by Csy4 in human cells. Tandem spacer containing premature crRNA (DR-spacer1-DR-spacer2-DR) was transcribed and processed by Csy4 into two mature crRNAs. White box: human U6 promoter (hU6); Gray box: direct repeats (DR); Red box: spacer 1; Blue box: spacer 2. **b** Quantitative PCR analysis of *HBB*, *HBG* and SOX2 transcription levels in HEK293T cells transfected with type I–F PaeCascade VPR (Csy3-VPR) with two crRNA expression vectors (crRNA1 + crRNA2) or customized CRISPR arrays (CRISPR arrays 1/2) targeting −100 bp (crRNA1) and −200 bp (crRNA2) upstream of TSS in Fig. 2. Upper: schematic illustration of Csy4 processing customized CRISPR arrays targeting two sites on the same gene. Lower: quantitative PCR analysis of *HBB*, *HBG*, and SOX2 transcription level in HEK293T cells. HEK293T cells were transfected with PaeCascade 2-vector systems (Csy3-VPR) and crRNA expression vectors as indicated. 48 h post-transfection, cells were lysed for RNA extraction and quantitative PCR assay. **c** Quantitative PCR analysis of multiplex gene activation in HEK293T cells transfected with type I–F PaeCascade VPR (Csy3-VPR) with 2 or 3 independent crRNA vectors or customized CRISPR arrays (CRISPR array) targeting different genes. Upper: Schematic illustration of Csy4 processing customized CRISPR arrays targeting two sites on different genes. Lower: quantitative PCR analysis of multiplex activating level in HEK293T cells. HEK293T cells were transfected with PaeCascade 2-vector systems (Csy3-VPR) and crRNA expression vectors as indicated. 2 crRNAs indicates two independent crRNA expression vectors. 3 crRNAs indicates three independent crRNA expression vectors. CRISPR array, customized CRISPR array in one vector. 48 h post-transfection, cells were lysed for RNA extraction and quantitative PCR assay. Ctrl: non-targeting crRNA control. Data represented three biological repeats and displayed as mean ± S.E.M. Statistical significance was calculated using one-way ANOVA (*$P < 0.05$; **$P < 0.01$; ***$P < 0.001$). Source data are provided as a Source Data file.

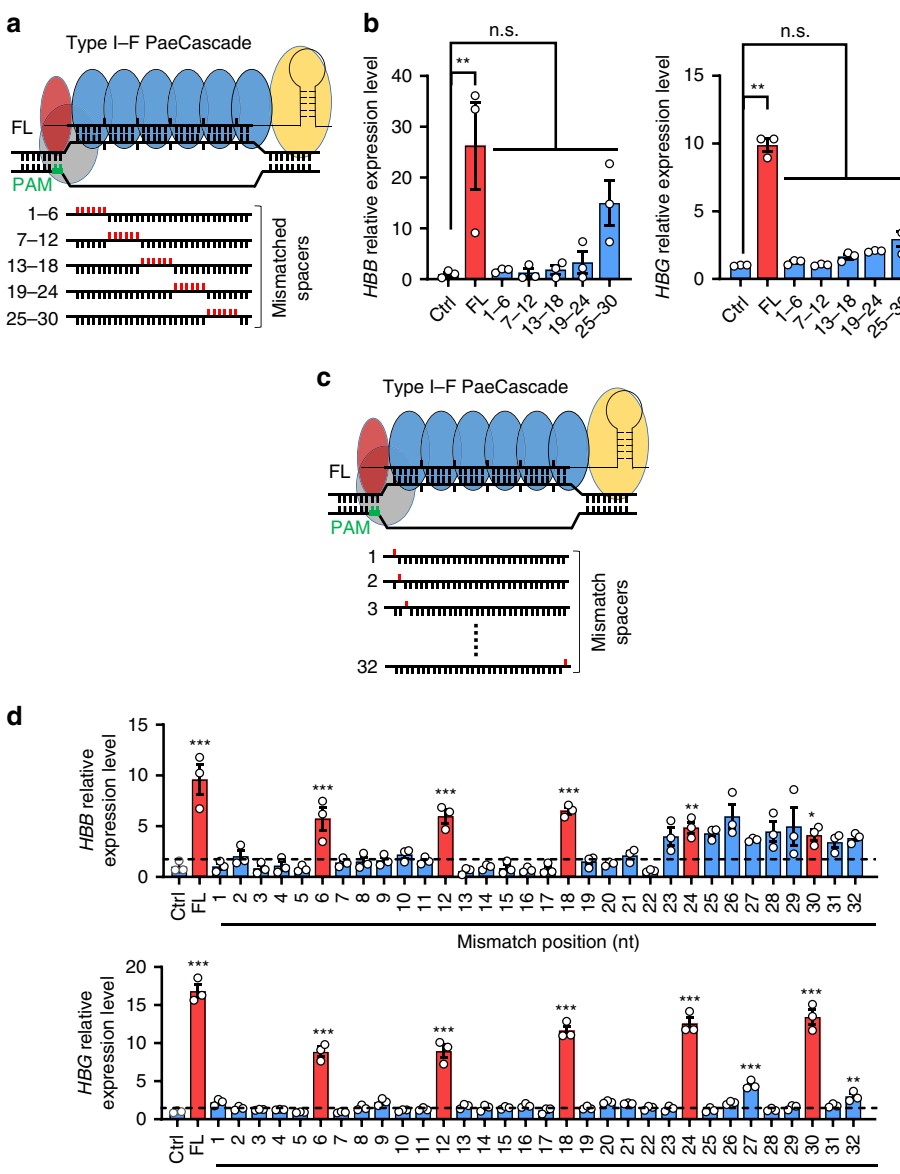

**Fig. 6 PaeCascade is sensitive to crRNA-DNA mismatch. a** Schematic illustration of crRNA variants containing 6-nt mismatches to the targeted site. There were five crRNA variants carrying 6-nt mismatches to the targeted DNA. Mismatched bases are highlighted in red and PAM is highlighted in green. Gray: Csy1; red: Csy2; blue: Csy3; yellow: Csy4. **b** Quantitative PCR analysis of *HBB* and *HBG* transcription levels in HEK293T cells transfected with type I–F PaeCascade VPR (Csy3-VPR) with full-length crRNA or 6-nt mismatched crRNA variants in (**a**). HEK293T cells were transfected with PaeCascade 2-vector systems (Csy3-VPR) and crRNA expression vectors. 48 h post-transfection, cells were lysed for RNA extraction and quantitative PCR assay. Ctrl: non-targeting crRNA control. **c** Schematic illustration of crRNA variants with single mismatches to the targeted site. There were 32 crRNA variants each carrying one single mismatch to the targeted DNA. Mismatched bases are highlighted in red and PAM is highlighted in green. **d** Quantitative PCR analysis of *HBB* and *HBG* transcription levels in HEK293T cells transfected with type I–F PaeCascade VPR (Csy3-VPR) with full-length crRNA or single nucleotide mismatched crRNA variants in (**c**). HEK293T cells were transfected with PaeCascade 2-vector systems (Csy3-VPR) and crRNA expression vectors. 48 h post-transfection, cells were lysed for RNA extraction and quantitative PCR assay. Ctrl: non-targeting crRNA control. Error bars represented three biological repeats and displayed as mean ± S.E.M. Statistical significance was calculated using one-way ANOVA (*$P < 0.05$; **$P < 0.01$; ***$P < 0.001$). Source data are provided as a Source Data file.

positions. Intriguingly, mismatches at every 6th position showed far less impact on PaeCascade-VPR activity, regardless of their distance to the PAM (Fig. 6d), consisting with its structure characteristic[53]. For type I–F Cascade, the binding of the target strand to crRNA follows a periodic 5 + 1 pattern[53]. The five consecutive base pairs followed by one base pair gap in which the unpaired nucleotides of crRNA and target strand kink out in opposite directions[53]. Therefore, the mismatches in per sixth nucleotide have less impact on target DNA binding and activation

efficiency. These data suggest that target binding by PaeCascade-VPR may be exceptionally sequence specific, with even residues far distal to the PAM playing a role in target DNA binding.

To further investigate the specificity of PaeCascade-VPR system, we searched for the target sites with overlapping target regions of PaeCascade-VPR and dCas9-VPR, which also had potential off-targets on the TSS of other genes (Fig. 7). To find out the off-target genes, we search for two groups of the potential off-target sites. We searched potential off-target sites with ≤4

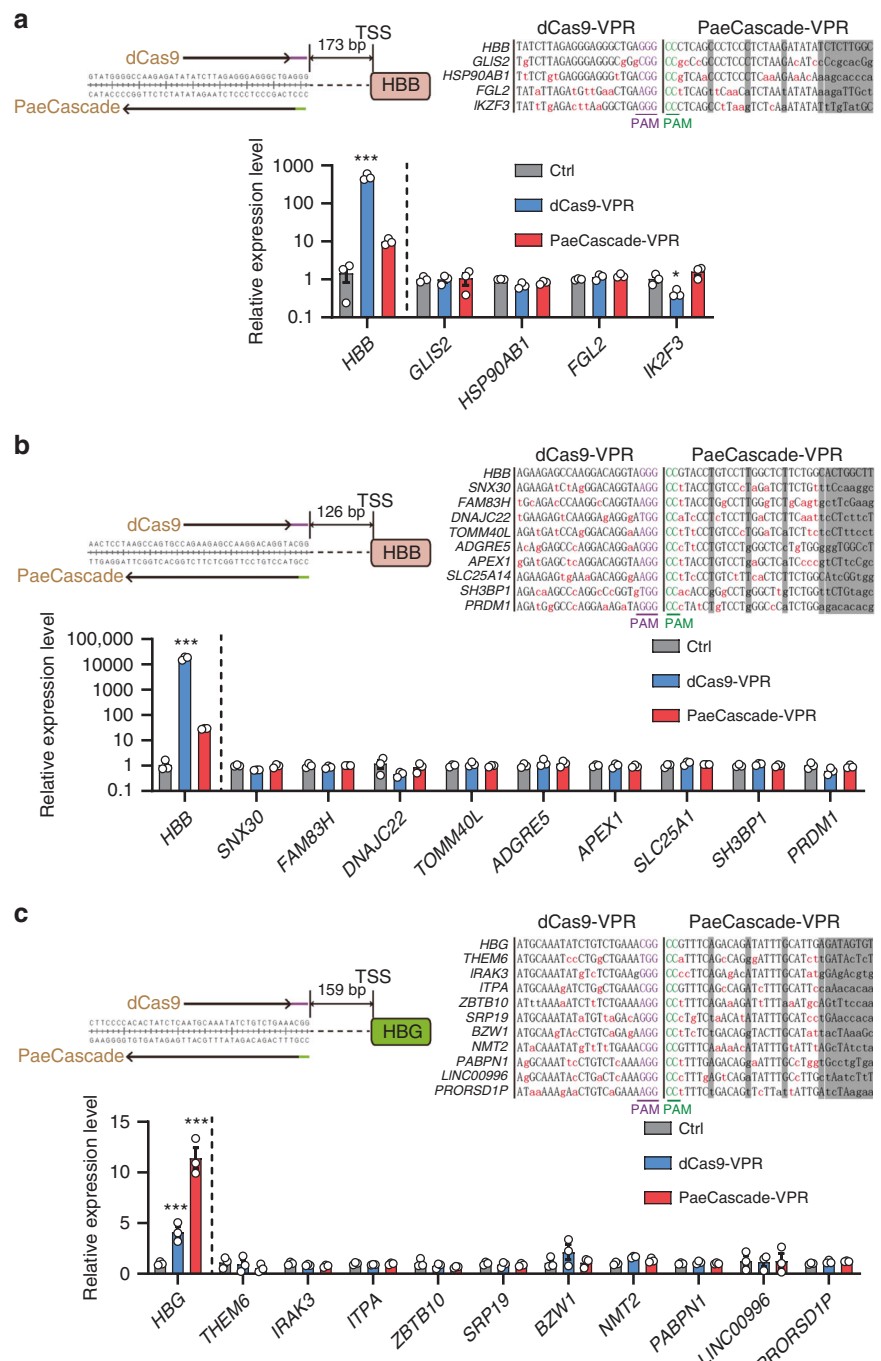

**Fig. 7 PaeCascade-VPR is highly specific. a** Off-target effects of gRNA and crRNA targeting the site that is 173 bp upstream the *HBB* TSS. Upper panel: schematic illustration of overlapping gRNA and crRNA designed. Potential off-target sites are shown on the right. Lower panel: quantitative PCR analysis of the mRNA level of *HBB* and the predicted off-target genes. **b** Off-target effects of gRNA and crRNA targeting the site that is 126 bp upstream the *HBB* TSS. Upper panel: schematic illustration of overlapping gRNA and crRNA designed. Potential off-target sites are shown on the right. Lower panel: quantitative PCR analysis of the mRNA level of *HBB* and the predicted off-target genes. **c** Off-target effects of gRNA and crRNA targeting the site that is 159 bp upstream the *HBG* TSS. Upper panel: schematic illustration of overlapping gRNA and crRNA designed. Potential off-target sites are shown on the right. Lower panel: quantitative PCR analysis of the mRNA level of *HBG* and the predicted off-target genes. On target sites and putative off-targets were listed. Mismatched bases were labeled in red. Purple: PAM sequence of dCas9-VPR. Green: PAM sequence of PaeCascade-VPR. Sequences in gray box indicated every six base and the 25–32th bases on PaeCascade targets, which were not counted as mismatches. Cells transfected with GFP were used as control. Error bars represented three biological repeats and displayed as mean ± S.E.M. Statistical significance was calculated using one-way ANOVA (*$P < 0.05$; ***$P < 0.001$). Source data are provided as a Source Data file.

mismatches to SpCas9 gRNA as the first group of putative off-targets. Taken the features of PaeCascade crRNA into consideration, mismatches on per 6th bases in PaeCascade crRNA had less impact on *HBB* and *HBG* activation (Fig. 6d), which may be tolerable for target binding. Also, mismatches on 25–32th bases were more tolerable than other bases (Fig. 6b, d). Previously studies also indicated that PAM-proximal region of type I CRISPR was more important for its binding capacity, and ≥5

mismatches would abolish type I CRISPR interference[59,60]. So we allowed mismatches in 6th, 12th, 18th, and 24–32th positions, and found all the possible off-targets with ≤4 mismatches to PaeCascade crRNA as the second group of putative off-targets. Then taking the two groups together, all the possible off-targets were predicted through sequence similarity, which must also lay on the promoter (≤2 kb upstream or downstream TSS) of a certain gene. According to the criteria above, we searched for target sites on *HBB* and *HBG* promoters. We found three regions with overlapping target sites of PaeCascade and dCas9 for off-target analysis (Fig. 7). The RNA level of *HBB* or *HBG* and their predicted off-target genes were then detected. With the crRNAs or gRNAs targeting to *HBB* 173 bp upstream TSS, *HBB* 126 bp upstream TSS or *HBG*, PaeCascade-VPR and dCas9-VPR can increase the transcription level of *HBB* and *HBG* as expected (Fig. 7). For both PaeCascade-VPR and dCas9-VPR, no off-target activations can be detected in all the putative off-target genes (Fig. 7). These results indicated that the type I–F PaeCascade-VPR system is comparable to dCas9-VPR and may have a high specificity as a transcription activator in human cells.

## Discussion

In this study, we demonstrated that the type I–F CRISPR–Cas system could be repurposed to activate endogenous gene expression in human cells. Fusing the Csy3 subunit of type I–F PaeCascade to transcription domain (VPR) led to a crRNA-dependent reporter and endogenous gene activation (Figs. 1, 2). And at most target genes, PaeCascade-VPR was much efficient than dCas9-VPR, dAsCas12a-VPR, and EcoCascade-VPR (Fig. 3b–e). Besides, having each Csy subunit expressed independently further improved activation efficiency (Fig. 1g). Moreover, compared to dCas9-VPR, the activation efficiency could be further improved by extending the spacer length of crRNA to recruit more Csy3-VPR protein to target genes (Fig. 4a). Customized CRISPR arrays enabled efficient multiplex gene activation in human cells (Fig. 5). Saturated mutation of crRNA spacer sequence revealed that target DNA binding by PaeCascade was sensitive to crRNA-DNA mismatch, suggesting that transcription activation by PaeCascade-VPR might be specific (Fig. 6d). And actually, we did not observe any off-target effects in the putative off-target genes of PaeCascade-VPR (Fig. 7). Taken together, these data prove that PaeCascade-VPR is a good programmable transcription activator in human cells.

We found that all subunits of PaeCascade (Csy1, Csy2, Csy3, and Csy4) could be fused with VPR without disturbing the formation of functional PaeCascade complex (Fig. 1g), which provides great flexibility on engineering. It is possible that we can activate gene expression with different kinds of effectors: Cascade-TET1 (Ten-Eleven Translocation dioxygenase1) fusion for DNA demethylation; Cascade-p300 fusion for histone acetylation; Cascade-VP64 or Cascade-VPR (VP64-p65-Rta) fusion for transcriptional factor recruitment, and achieve stronger and more persisted gene activation through combining these three methods properly[61–64]. So, it might be possible to fuse more transcription regulating domains to the PaeCascade complex to improve activation efficiency or even achieve long term memory activating of endogenous genes. While our manuscript was under preparation, Adrian et al. reported transcription regulation by type I–B and type I–E CRISPR–Cas system in human cells[65]. Although type I–B also used four subunits to activate endogenous genes, type I–B tool was not better than dCas9. Furthermore, type I–B Cas7 (Csy3 equivalent) failed to induce gene activation when fused to transcription activator[65]. However, transcription activator fused to Csy3 subunit of type I–F CRISPR system showed the highest activating level (Fig. 1g). It was even better than dCas9

and other transcription activation systems at most (4/5) tested endogenous sites. In addition to gene activation, PaeCascade subunits might be fused with transcription repressor to repress gene expression, or nuclease domain to cleave target DNA in human genome[65].

Previous studies of type I CRISPR have identified an eight nucleotide PAM-proximal seed region (1–5th, 7th, 8th bases)[26,59,60], and the imprecise base-pairing at every sixth position within the 32 nucleotide crRNA sequence[53,54,66], owing to structure feature of every sixth base being flipped out of the RNA–DNA duplex upon target binding. Being consistent with these studies, we found that the PAM-proximal position is crucial for gene activation of PaeCascade-VPR (Fig. 6a). In contrast, every sixth base had a relatively weak influence on its binding (Fig. 6d). Recent studies that generating long-range deletions in human embryonic stem cells or HEK293T with EcoCascade-Cas3 revealed no prominent off-target effect either by deep sequencing or by whole genome sequencing[33,34]. It had been shown that type I–B and type I–E CRISPR–Cas could induce specific targeted transcription activation in human cells without crRNA-dependent off-target effects[35]. According to our research data, we could achieve a high transcription activation level without activation of putative off-target genes by type I–F PaeCascade (Fig. 7). These data indicate that the specificity of type I system is high in mammalian cells.

Transcription activation could be used to upregulate therapeutic gene expression. For example, activating *HBB* or *HBG* gene expression might be used to treat β-thalassemia. Further studies are needed to investigate the function and the delivery of PaeCascade-VPR in primary cells (e.g., hematopoietic stem cell) or in vivo. Other aspects, including the cytotoxicity and immunogenicity of type I–F system, should be studied in detail. Further efforts improving the activation efficiency of PaeCascade-VPR are also important as well. Only then can type I–F PaeCascade-VPR be a tool for therapeutic gene expression activation. In brief, we found that PaeCascade-VPR can induce targeted gene activation without off-target effects, indicating that PaeCascade-VPR is a good programmable transcription activator in human cells. Regulating of gene expression by Type I–F CRISPR system broadens the usage of CRISPR system as a gene regulating tools in mammalian cells.

## Methods

**Cell culture**. HEK293T cells were obtained from ATCC and cultured in Dulbecco's modified Eagle medium (Corning, 10-013-CVR) supplemented with 10% fetal bovine serum at 37 °C and 5% $CO_2$ in humidified incubator, with daily medium change. Cells were split every 2–3 days. All the cells were mycoplasma negative. Transient transfection of HEK293T cells was performed using PEI (Polysciences, 24765-1). Cells were lysed by Trizol 48 h later for qPCR analysis or collected 72 h later for flow cytometry analysis.

**Plasmids and vectors**. Type I–F Cascade (from *Pseudomonas aeruginosa*) *E. coli* expression plasmids were obtained from Addgene (pCsy_complex, 89232). Type I–Fv (from *Shewanella putrefaciens*) Cas7fv, Cas5fv, Cas6fv cassettes were cloned into the pET28a vector (Sigma-Aldrich, 69864-3CN) as a polycistronic operon and include an N-terminal His-tagged Cas7fv fusion (pET28-type I–Fv). The crRNA sequence was cloned into pACYC184 (NEB, X06403) for bacterial expression. Condon-optimized Cas subunits were sub-cloned into px601 (Addgene, #61591) (replacing the SaCas9 gene) for transfection into mammalian cells. A site for spacer cloning flanked by two Csy4 direct repeats (DR) or Cas6f direct repeats was ligated into lentiGuide-Puro (addgene #52963) between BsmBI and EcoRI restriction sites to generate pLenti-crRNA-IF or pLenti-crRNA-IFv vectors. Oligos containing spacer sequences were annealed and ligated into pLenti-crRNA-IF or pLenti-crRNA-IFv for crRNA expression in mammalian cells. For spacer mutant crRNA cloning, oligos with various of mutant spacer were annealed and ligated into pLenti-crRNA-IF. Sequences are listed in Supplementary Data 1–4. Sequences of plasmids for expression of PaeCascade-VPR, including pCsy1-Csy2, pCsy3-VPR-Csy4, and pCsy-crRNA-EV, are listed in Supplementary Data 5.

**Protein expression and purification**. Type I–F and type I–Fv Cascade were expressed and purified using prokaryotic systems. Briefly, BL21 Star™ (DE3) *E. coli* cells (Thermo Fisher) were transformed with pCsy_complex (or pET28-type I–Fv) together with pACYC184 vector containing corresponding crRNA. When $OD_{600}$ reached 0.6, protein expression was induced by 5 mM IPTG and cultured for another 12 h at 25 °C. Cells were harvested and suspended in buffer A (20 mM HEPES-Na pH 8.0, 250 mM NaCl, 20 mM KCl, 20 mM $MgCl_2$, 40 mM imidazole), disrupted by sonication and purified using Ni Sepharose 6FF column (GE Healthcare). Size exclusion chromatography was performed on a Superdex 200 Tricon 10/300 column (GE Healthcare) in buffer B (20 mM HEPES-Na pH 7.0, 150 mM NaCl, 1 mM DTT, 1 mM EDTA). Fractions containing the target complex were collected. Protein concentration was measured by BCA protein assay kit (Thermo Fisher, 23225).

**Electrophoresis mobility shift assay (EMSA)**. Target oligonucleotides used were detailed in Supplementary Data 6. Substrate dsDNA was prepared by annealing two complementary oligos with a molar ratio of 1:1. 200 nM of substrate DNA were incubated with various amount of purified recombinant protein complex at 37 °C for 30 min in binding buffer (50 mM HEPES-Na pH 7.0, 50 mM NaCl, 1 mM DTT, 1 mM EDTA, 10 IU RNase inhibitor (Thermo Fisher, EO0381)). The products were then separated via non-denaturing TBE-PAGE and stained by Gel-red™ (Biotium, 41000).

**Quantitative PCR (qPCR)**. Briefly, total RNA was extracted by TRIZOL (Thermo Fisher) following the manufacture's instruction and quantified by Nanodrop 1000 (Thermo Fisher). The reverse transcription was carried out using the Prime-Script™RT reagent Kit (TAKARA, RR047Q) following the manufacture's instruction. Quantitative PCR was carried out in qTOWER[3] system (Analytikjena) using TAKARA TB Green II Real-Time PCR Master Mix following the manufacture's instruction. Quantitative PCR was performed with indicated primer for specific genes, and *GAPDH* served as control. The relative expression level was determined by $-\Delta\Delta Ct$ method. qPCR primers are listed in Supplementary Data 7.

**Flow cytometry analysis**. Cell was digested by 0.25% trypsin, and then trypsin digestion was terminated by DMEM containing 10% FBS. Cells were collected and suspended in PBS. The GFP positive cells were detected by CytoFLEX (Beckman).

**Western blot (WB)**. Three days post-transfection, cells were lysed in RIPA buffer with protease inhibitor cocktail. Samples were centrifuged at $14,000 \times g$ for 10 min. The supernatant was harvested and quantified using BCA protein assay kit (Thermo Fisher, 23225) on Victor X5. 25 μg protein was mixed and boiled with 5 × SDS loading buffer. Samples were separated using SDS-PAGE assay. Protein was transferred to nitrocellulose membranes (Bio-Rad) for 1 hour in transfer buffer at 300 mA. The membranes were blocked at room temperature for 20 min in 5% milk-TBST and incubated with the primary antibody in 3% BSA-TBST at RT for two hours. Then the membranes were washed in TBST and incubated with secondary antibody in 3% BSA-TBST at RT for one hour and washed in TBST. Blots were visualized using Odyssey finally. The antibodies used for WB were listed below. Rabbit polyclonal anti-GAPDH (Abmart, P30008M) (1:5000 dilution), mouse monoclonal anti-HA antibody (Sigma, H9658) (1:5000 dilution), goat anti-rabbit secondary antibody (Odyssey, 926-32211) (1:5,000 dilution) and the goat anti-mouse secondary antibody (Odyssey, 926-68070) (1:5,000 dilution).

**Off-target perdition**. To predict the putative off-targets for dCas9-VPR, we first searched off-targets with ≤4 mismatches to SpCas9 gRNA. And for the prediction of PasCascade-VPR, we allowed mismatches in 6th, 12th, 18th, and 24–32nd position, and found all the possible off-targets with ≤4 mismatches to PaeCascade crRNA. Then all the possible off-target sites were predicted through sequence similarity, which also lay on the promoter (≤2 kb) of a certain gene (UCSC, with Integrated Regulation from ENCODE Tracks and GeneHancer Regulatory Elements and Gene Interactions). Sequences of all putative off-targets were listed in Supplementary Data 8.

**Significant analysis**. All data were processed and tested using GraphPad Prism 7.0. For all the data, Gaussian distribution was detected by Shapiro–Wilk normality test. One-way ANOVA (for data having more than two groups) or unpaired *t* test (for data having only two groups) was used for data with Gaussian distribution (Normal distribution) and equal SDs. Otherwise, Kruskal–Wallis or Mann–Whitney test was used. Data were displayed as mean ± S.E.M. Statistical significance level: n.s., not significant; *$P < 0.05$; **$P < 0.01$; ***$P < 0.001$.

**Reporting summary**. Further information on research design is available in the Nature Research Reporting Summary linked to this article.

## Data availability

All relevant data are available upon request. Sequences of plasmids for expression of PaeCascade-VPR, including pCsy1-Csy2, pCsy3-VPR-Csy4, and pCsy-crRNA-EV, are listed in Supplementary Data 5. The source data for Figs. 1b, c, e, f, g, 2a, b, d, 3b, c, d, e, 4, 5b, c, 6b, d, 7 and Supplementary Figs. 2, 3b, 4b, and 7 are provided as a Source Data file.

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

## Acknowledgements

We would like to thank all the members in Zhou Songyang's lab for discussions and supports. This work was supported by the National Key R&D Program of China (2017YFA0102801, 2017YFC1001901, and 2017YFC1001603), the National Natural Science Foundation (91640119, 91019020, 81330055, 31671540, and 31601196), the Guangdong Special Support Program (2019BT02Y276), the Natural Science Foundation of Guangdong Province (2016A030310206 and 2014A030312011), the Guangzhou Science and Technology Project (201707010085 and 201803010020).

## Author contributions

Z.S., P.L., and Y.C. designed the experiments. D.L. helped with the paper. J.L., Y.C., S.Z., and Q.Z. performed the experiments. Z.S. and P.L. supervised the research. All authors discussed the results and commented on the paper.

## Competing interests

The authors declare no competing interests.
