## [Peer Review File · Nature Communications]

Reviewers' Comments:

Reviewer #1:

Remarks to the Author:

Chen et al. present a study of transcriptional control in human cells using Type I CRISPR-Cas systems. Specifically, the authors show that the *Pseudomonas aeruginosa* Type I-F Cascade machinery fused with a transcriptional activator and complexed with a targeting crRNA can be used to increase transcriptional levels in HEK293 cells. In particular, Csy3-VPR fusions enable varying levels of activation of 6 distinct genes (e.g. HBB, HGB, IL1R2 and others) by directing Cascade to various regions upstream of the TSS using different crRNA guides.

Though this is a timely topic and rising technology, the authors fail to present and discuss their results in light of a series of recently published papers (N=4) that illustrate how Type I CRISPR-Cas systems can be exploited to control transcription in various settings and organisms. While the authors do provide data showing that various levels of transcription can be achieved using PaeCascade, the novelty of the method and impact of the narrative are limited in light of (at least) four recently published papers (Cameron et al., 2019; Dolan et al., 2019; Pickar-Oliver et al., 2019; Young et al., 2019), that show how various Type I systems from various subtypes can achieve a broad spectrum of transcriptome and genome manipulation in various complex eukaryotes encompassing human and plant cells. In particular, the key results shown by the authors have all individually been reported in the past, notably with regards to:

- Using Class 1 systems to control eukaryotic transcription
- Fusing Cascade to transcriptional activators
- Elongating the crRNA: Cascade complex
- Achieving multiplexed editing using CRISPR arrays
- Identifying mismatch tolerance across crRNA:target DNA complexes

These 4 manuscripts, and possibly more as well as a few additional studies presumably in press strongly negatively impact the novelty and impact of the narrative in general and results presented in particular. In the abstract and introduction, the authors frame their narrative to state that "Class 1 systems could be used to control mammalian gene expression and edit (the) genome", but nearly a half dozen papers already showed this.

In several instances, the authors mis-characterize and incorrectly conceptualize actual CRISPRs (clustered regularly interspaced short palindromic repeats loci that contain several CRISPR repeats flanking CRISPR spacers). A CRISPR locus with 3 repeats and 2 spacers is not a "tandem crRNA", but rather an actual CRISPR. Multiplexed targeting using an actual CRISPR is not engineering per se.

For the manuscript to conclusively state that PaeCascade is better than members of the incumbent CRISPR toolbox, the authors must provide comparative insights vs. Cas9, Cas12 and other previously used Cascade(s). Also, the varying and relative extents of alteration of transcription levels must be more thoroughly discussed and presented in light of all the previously published relevant studies.

Given the widespread concerns about the CRISPR toolbox specificity and efficiency, the authors should and perhaps must provide specificity results and also compare and contrast these to widely used and relevant controls (at least SpyCas9 and possibly EcoCascade). This should be done for at least 2 or 3 of their target transcripts, and ensure that the targeting spacers are (at least partially) overlapping.

When the authors present an argument for "tunable transcriptional activation", they fail to explain how guides can be selected or designed to specifically achieve a particular level of transcriptional control. Rather, it seems variability in targeting efficiency between guides can be used with varying levels of success to alter transcription levels across a dynamic range. This is arguably more "unpredictably

variable" than it is "tunable" per se, and thus the authors' narrative is (at least partially) incorrectly framed.

Graphically, the figures provided fall short of the journal standards and pale in comparison to displays published in manuscripts discussing Cascade-based repression, notably with regards to cartoons (Cascade should be consistently represented using a more appropriate rendition of the various elements involved), sequence details (guide RNA sequences and target sequences with PAMs should all be specified and displayed) and standards (markers and ladders on gels, + and - controls on graphs) and comparisons (vs. dCas9 and possibly dCas12 or perhaps other Cascade tools). Also, the authors must provide the PaeType I-F and SpuType I-Fv "natural" operons in their original host, and correctly name all cas genes using the latest nomenclature standards. All like-material should be represented using a single cartoon display (there are varying sizes and color schemes for the same elements in figures 1,2,3,4, and 5, such as Cascade and guides).

From a literature citation and narrative framing standpoint, there are multiple categories of previous studies that must be cited and discussed, with regards to Cascade use for transcriptional control in eukaryotes; CRISPR-Cas system biology and crRNA processing, Type I systems (notably Pae Type I-F), and possibly more. Indeed, besides the aforementioned shortcomings regarding the published studies illustrating the potential of Type I systems for transcriptional control, the authors must provide more background and details and discussion on Type I CRISPR-Cas systems in general and Cascade and crRNA processing in particular. After all, it was first shown over a decade ago that CRISPR is driven by crRNA biogenesis by a Type I paper conspicuously missing from the cited references. Besides Type I-E systems, many early and also recent studies characterizing Type I-F CRISPR-Cas systems are missing, and it seems that PaeCascade has been more studied than the TfuCascade, as opposed to what the authors claim. This also applies to the section investing the impact of mismatch at periodic intervals along the crRN:targetDNA duplex.

From a narrative standpoint, there are grammatical issues throughout the manuscript that warrant extensive narrative editorial corrections and improvements (see abstract for examples of missing words and grammatical shortcomings).

Reviewer #2:

Remarks to the Author:

Summary

Chen et al. report the development of a programmable CRISPRa system for gene activation using a Type I-F CRISPR-Cas system. Most reported implementations of CRISPRa are derived from class 2 systems. Type I systems present challenges because the RNA-protein complex that binds DNA includes multiple protein components, although there are potential advantages for recruiting multiple effectors via distinct subunits. The authors fused the VPR activator to various subunits of the Type I-F Cascade complex and demonstrate transcriptional activation at a heterologous reporter gene and multiple endogenous genes. They assess the effects of varying crRNA position, multiplexing crRNAs, varying crRNA spacer length, and crRNA-DNA mismatches.

A comparable Type I system was recently described in Nature Biotechnology (Pickar-Oliver et al., 2019 doi: 10.1038/s41587-019-0235-7; cited by the authors as ref 28). This paper demonstrated both transcriptional activation and repression, and included extensive genome-wide experiments to assess the specificity of the system. For transcriptional activation, this work used the p300 activator, although they also showed a comparison with the VPR activator. There is significant value in publishing multiple

implementations of new CRISPRa systems; several Type II CRISPRa systems were published in prominent journals in 2013-2015.

The manuscript under review reports a set of unique and distinct findings that will be a valuable contribution to the literature. Most importantly they demonstrate that a Csy3-VPR fusion is an effective activator; Pickar-Oliver et al. did not report testing the analogous Cas7(Csy3)-p300 fusion. This finding is significant because multiple copies of Csy3 bind the crRNA spacer. By increasing the spacer length, the authors were able to increase gene activation, presumably by recruiting more copies of Csy3-VPR.

I have several comments that the authors should consider.

Major Comments

1. The authors should comment on the specificity of their system. While I do not think it is absolutely necessary to perform genome-wide sequence experiments (as in Pickar-Oliver et al.), the authors should include some gene activation experiments with off-target crRNAs. They should also briefly discuss the available literature on the specificity of Type I CRISPR-Cas systems.

2. On pg 8, the authors describe a two-plasmid delivery strategy for the Cascade complex proteins. The logic here is confusing on multiple levels. The motivation is that Csy3 is present in multiple copies and so might be limiting if delivered in the same plasmid as the other components. Based on this logic, it seems like the authors should deliver Csy3 on one plasmid and the other three proteins on a second plasmid, and then increase the ratio of Csy3 plasmid to the other plasmid. Instead, the authors delivered the proteins on two plasmids in pairs (Csy1/2 and Csy3/4), and they did not describe whether they varied the ratios of these two plasmids for transfections. Given the system described, it is unclear why there was any benefit to the helper strategy versus the 4-component single plasmid strategy. There is no explanation for how the helper plasmid strategy enabled the authors to change the ratio of Csy3 to the other components. Some additional experiments to address this point would be useful.

3. At multiple points in the manuscript, the authors suggest that an advantage of the Type I system is that multiple subunits will facilitate novel applications by allowing multiple different tags to be recruited to the same complex. There are no experiments that justify this claim, although I agree that it is possible. In fairness, Pickar-Oliver et al. made the same claim, also without experimental justification. The authors could consider specifying exactly what types of multiple effector recruitment schemes might be desirable.

4. On pg 9, the authors compare their system to dCas9-VPR and suggest that their system is generally more effective. CRISPRa output can vary with sgRNA sequence and position relative to the TSS. It seems difficult to make a blanket conclusion about dCas9-VPR vs. the Type I-F system with the data shown. How did the authors choose the target sites for dCas9-VPR? Do they know if they are comparing the most effective sites between each system?

Minor comments

1. The introduction could be much more concise. There is more background description of class 1 and class 2 CRISPR systems than is really needed. Conversely, some details that might be helpful are missing. For example, it would be useful to explain that the Type I Cascade complex recruits a Cas3 nuclease, and that the system described here does not include the nuclease, just the DNA binding components of the Cascade complex.

2. In the last sentence of the abstract and in the last paragraph of the discussion, the authors suggest that this CRISPRa system could be useful in therapeutic applications. There are significant challenges for

the use of any CRISPRa system as a therapeutic, including immunogenicity, delivery, and off-target effects. The authors do not indicate whether the Type I system described here might have any unique advantages compared to other CRISPRa systems. It is unclear why the authors are making this claim and whether it is justified by the specific findings reported in this paper.

3. On pg 3, the authors write: "Comparing to current widely used class 2 systems, the multiple-subunit type I CRISPR has distinct property in genome editing¹⁵, and the multiple subunit arrangement in the cascade raises the possibility of refining type I CRISPR into a "Swiss Army Knife" like platform for gene engineering." At this point in the manuscript, this statement is vague and confusing. The authors do a better job explaining their point on pg 4.

5. On pg 4, the authors write: "In contrast to Type I-E and I-B, *Pseudomonas aeruginosa* type I-F and *Shewanella putrefaciens* type I-Fv systems have fewer subunits and also bind dsDNA in bacteria." This statement is confusing. It reads as if the authors are saying I-E and I-B systems do not bind dsDNA, which is unlikely to be the point they are trying to make.

6. On pg 5, the authors write: "Comparing to current single effector class 2 systems, the multiple subunits in type I-F and type I-Fv might provide more combinations for tagging and increase signal strength if genetic modulators or tags are fused to different subunits. These characteristics and potential advantages make them better candidates as novel gene editing and regulating tools." The first sentence is plausible but the second sentence is speculation. There are many factors that could be considered to assess whether a system is "better". Also, note that here the authors are implying that having more subunits is better, but their justification for using I-F instead of I-E was that I-F has fewer subunits (pg. 4).

7. In Figure 1a, there are two arrows pointing to bands on the gel. The lower arrow is not described in the figure legend and does not seem to point to anything of significance.

8. In Figure 1b, where is the unshifted ssDNA on the gel?

9. In Figure 1b, why is one of the samples here non-target ssDNA? In panel a, the corresponding sample is non-target dsDNA.

10. On page 9, the authors attempt to activate multiple endogenous genes. CRISPRa output can vary depending on the basal expression level of the target gene (e.g. doi:10.1038/nmeth.3871 & doi:10.1038/nature14136). Are similar trends occurring here? Can the authors offer any interpretation for why different outputs were observed with different endogenous gene targets?

11. On pg 11, the authors write: "The ability of PaeCascade-VPR to activate multiplex genes simultaneously with a single crRNA array instead of using individual crRNAs enormously simplifies the activation system, increasing the transfection efficiency and removing the roadblock of multiplex activation using type II CRISPR system." This statement is far too broad. There are many multiplex sgRNA delivery strategies that have been reported for Class 2 Type II CRISPR systems.

12. On pg 12, the authors describe how mismatches every 6th position are tolerated. This finding is interesting. Can the authors discuss further and provide any interpretation? Is there any precedent or structural rationale for this finding?

13. On pg 14, the authors write: "Furthermore, type I-B Cas7 (Csy3 equivalent) failed to induce gene activation when fused to transcription activator (28)." As best I can tell, ref 28 (Pickar-Oliver et al.) did not report testing a Cas7-p300 fusion. Can the authors clarify?

MS title: Repurposing Type I-F CRISPR-Cas System as a Transcriptional Activation Tool in Human Cells

MS number: NCOMMS-19-37409

We would like to thank the reviewers for their useful and considerable comments and suggestions. We have performed additional experiments and revised our manuscript accordingly. Please find below our point-by-point response.

Reviewer #1:

Chen et al. present a study of transcriptional control in human cells using Type I CRISPR-Cas systems. Specifically, the authors show that the *Pseudomonas aeruginosa* Type I-F Cascade machinery fused with a transcriptional activator and complexed with a targeting crRNA can be used to increase transcriptional levels in HEK293 cells. In particular, Csy3-VPR fusions enable varying levels of activation of 6 distinct genes (e.g. HBB, HGB, IL1R2 and others) by directing Cascade to various regions upstream of the TSS using different crRNA guides.

1. Though this is a timely topic and rising technology, the authors fail to present and discuss their results in light of a series of recently published papers (N=4) that illustrate how Type I CRISPR-Cas systems can be exploited to control transcription in various settings and organisms. While the authors do provide data showing that various levels of transcription can be achieved using PaeCascade, the novelty of the method and impact of the narrative are limited in light of (at least) four recently published papers (Cameron et al., 2019; Dolan et al., 2019; Pickar-Oliver et al., 2019; Young et al., 2019), that show how various Type I systems from various subtypes can achieve a broad spectrum of transcriptome and genome manipulation in various complex eukaryotes encompassing human and plant cells. In particular, the key results shown by the authors have all individually been reported in the past, notably with regards to:

- Using Class 1 systems to control eukaryotic transcription
- Fusing Cascade to transcriptional activators
- Elongating the crRNA:Cascade complex
- Achieving multiplexed editing using CRISPR arrays
- Identifying mismatch tolerance across crRNA:target DNA complexes.

These 4 manuscripts, and possibly more as well as a few additional studies presumably in press strongly negatively impact the novelty and impact of the narrative in general and results presented in particular. In the abstract and introduction, the authors frame their narrative to state that "Class 1 systems could be used to control mammalian gene expression and edit (the) genome", but nearly a half dozen papers already showed this.

Answer: It is true that there are many papers already showed the usage of Class 1 system (type I-E and Type I-B) for controlling mammalian gene expression and editing genome ^{1, 2, 3, 4, 5}. However up to now, class 1 type I-F CRISPR has not been reported to be used in mammalian cells. Importantly, we have shown some special features of type I-F PaeCascade in this study. The type I-F PaeCascade is simpler than type I-E and type I-B. In addition, our

PaeCascade-VPR system can activate gene expression by multiple Csy3 (Cas7)-VPR fusion, which enhanced the transcription activating level in a dosage-dependent manner (**new Fig. 1g**). In contrast, the Cas7 (from either type I-E or type I-B) fused with VPR or P300 could not activate gene expression³. The activation level can be increased further through lengthening the crRNA spacer sequence through recruiting more Csy3-VPR for transcription activation, which has not been reported (**new Fig. 4a**). We compared the transcription activation efficiency of PaeCascade-VPR, dCas9-VPR, dAsCas12a-VPR, and EcoCascade-VPR, and we found that PaeCascade-VPR had high activity in most sites (4/5) (**new Fig. 3**), indicating that PaeCascade-VPR would be a good programmable transcription activator in human cells.

2. In several instances, the authors mis-characterize and incorrectly conceptualize actual CRISPRs (clustered regularly interspaced short palindromic repeats loci that contain several CRISPR repeats flanking CRISPR spacers). A CRISPR locus with 3 repeats and 2 spacers is not a "tandem crRNA", but rather an actual CRISPR. Multiplexed targeting using an actual CRISPR is not engineering per se.

Answer: We thank the reviewer for pointing out our inaccurate description. Referring to the paper published⁶, we have changed the description of "tandem crRNA" to "customized CRISPR arrays" in the manuscript.

3. For the manuscript to conclusively state that PaeCascade is better than members of the incumbent CRISPR toolbox, the authors must provide comparative insights vs. Cas9, Cas12 and other previously used Cascade(s). Also, the varying and relative extents of alteration of transcription levels must be more thoroughly discussed and presented in light of all the previously published relevant studies.

Answer: We apologize for not explaining why PaeCascade is better clearly. On the one hand, Csy4 of type I-F CRISPR has the pre-crRNA processing ability, which enables multiple crRNAs to be expressed in the form of customized CRISPR array instead of being driven by independent promoters. It simplifies the multiplex genes activation strategy to some extent. On the other hand, the type I-F Cascade, consisting of four subunits, is easier to be manipulated as a CRISPR tool than the recently used type I-E Cascade, which consists of five subunits.

As suggested, we compared the activation efficiency of dCas9, dAsCas12a, EcoCascade and our PaeCascade fused with VPR as shown in **new Fig. 3** (Please see Fig. 1 below). And we designed partially overlapping gRNAs or crRNAs upstream four endogenous genes (*SOX2*, *IL1B*, *HBG*, and *HBB*). PaeCascade-VPR showed higher transcription activation levels for *HBG*, *SOX2* and *IL1B* than other VPR-based systems. Cas9 and PaeCascade had similar activation level when targeting -244bp region upstream *HBB* TSS. These data supported the notion that PaeCascade-VPR might be more efficient than other VPR-based CRISPR transcription activation systems at least for the genes we tested.

Fig. 1. Transcription activation by Type I-F PaeCascade-based transcription activator and other VPR-based systems

(a) Schematic illustrating designed gRNAs and crRNAs of dCas9, dAsCas12a, EcoCascade, and PaeCascade targeting different regions upstream the transcriptional start site (TSS) of four genes. Arrow directions indicated the 5' to 3' sequence of the spacer. Colored lines (purple, green, yellow, and blue) represented PAM motif.

(b) Quantitative PCR analysis of HBB transcription levels in HEK293T cells transfected with dCas9-VPR, dAsCas12a-VPR, EcoCascade-VPR, or PaeCascade VPR (Csy3-VPR).

(c) Quantitative PCR analysis of HBG transcription levels in HEK293T cells transfected with dCas9-VPR, dAsCas12a-VPR, EcoCascade-VPR, or PaeCascade VPR (Csy3-VPR).

(d) Quantitative PCR analysis of SOX2 transcription levels in HEK293T cells transfected with dCas9-VPR, dAsCas12a-VPR, EcoCascade-VPR, or PaeCascade VPR (Csy3-VPR).

(e) Quantitative PCR analysis of IL1B transcription levels in HEK293T cells transfected with dCas9-VPR, dAsCas12a-VPR, EcoCascade-VPR, or PaeCascade VPR (Csy3-VPR). Ctrl: non-targeting crRNA control. Data represented three biological repeats and displayed as mean \pm S.E.M. Statistical significance was calculated using one-way ANOVA (*, $P < 0.05$; ***, $P < 0.001$).

4. Given the widespread concerns about the CRISPR toolbox specificity and efficiency, the authors should and perhaps must provide specificity results and also compare and contrast these to widely used and relevant controls (at least SpyCas9 and possibly EcoCascade). This should be done for at least 2 or 3 of their target transcripts, and ensure that the targeting spacers are (at least partially) overlapping.

Answer: We thank the reviewer for the helpful suggestion. We have performed the experiment as suggested. We have designed three overlapping target regions of SpCas9 and PaeCascade. We picked all the possible off-target genes (≤ 4 base mismatches to either Cas9 gRNA or PaeCascade crRNA) to test the specificity of these two systems. As shown in **new Fig. 7** (Please see Fig. 2 below), we found both dCas9-VPR and PaeCascade-VPR could activate the on-target genes efficiently, while none of them led to the unwanted activation of putative off-target genes. These results indicated that the specificity of PaeCascade is comparable to that of SpCas9.

Fig. 2. PaeCascade-VPR is highly specific.

(a) Off-target effects of gRNA and crRNA targeting the site that is 173bp upstream the HBB TSS. Upper panel: Schematic illustration of overlapping gRNA and crRNA designed. Potential off-target sites are shown on the right. Downer panel: Quantitative PCR analysis of the mRNA level of HBB and the predicted off-target genes.

(b) Off-target effects of gRNA and crRNA targeting the site that is 126bp upstream the HBB TSS. Upper panel: Schematic illustration of overlapping gRNA and crRNA designed. Potential off-target sites are shown on the right. Downer panel: Quantitative PCR analysis of the mRNA level of HBB and the predicted off-target genes.

(c) Off-target effects of gRNA and crRNA targeting the site that is 159bp upstream the HBG TSS. Upper panel: Schematic illustration of overlapping gRNA and crRNA designed. Potential off-target sites are shown on the right. Downer panel: Quantitative PCR analysis of the mRNA level of HBG and the predicted off-target genes. On target sites and putative off-targets were listed. Mismatched bases were labelled in red. Purple: PAM sequence of dCas9-VPR. Green: PAM sequence of PaeCascade-VPR. Sequences in grey box indicated every six base and the 25-32th bases on PaeCascade targets, which were not counted as mismatches. Cells transfected with GFP were used as control. Error bars represented three biological repeats and displayed as mean \pm S.E.M. Statistical significance was calculated using one-way ANOVA (***, $P < 0.001$).

5. When the authors present an argument for "tunable transcriptional activation", they fail to explain how guides can be selected or designed to specifically achieve a particular level of transcriptional control. Rather, it seems variability in targeting efficiency between guides can be used with varying levels of success to alter transcription levels across a dynamic range. This is arguably more "unpredictably variable" than it is "tunable" per se, and thus the authors' narrative is (at least partially) incorrectly framed.

Answer: We apologize for the argument because of our confusing description. As the reviewer indicated, it is variable in targeting efficiency between guides. So we removed the word "tunable" and corrected our narrative as suggested.

6. Graphically, the figures provided fall short of the journal standards and pale in comparison to displays published in manuscripts discussing Cascade-based repression, notably with regards to cartoons (Cascade should be consistently represented using a more appropriate rendition of the various elements involved), sequence details (guide RNA sequences and target sequences with PAMs should all be specified and displayed) and standards (markers and ladders on gels, + and – controls on graphs) and comparisons (vs. dCas9 and possibly dCas12 or perhaps other Cascade tools). Also, the authors must provide the PaeType I-F and SpuType I-Fv "natural" operons in their original host, and correctly name all cas genes using the latest nomenclature standards. All like-material should be represented using a single cartoon display (there are varying sizes and color schemes for the same elements in figures 1,2,3,4, and 5, such as Cascade and guides).

Answer: We thank the reviewer for pointing out our mistake in detail. As shown in the new figures, we have corrected the cartoons, data presenting standards and added more sequence details, Pae Type I-F and Spu Type-I-Fv "natural" operons in the original host in **new Fig. 1a** as suggested. Also, the names of all *cas* genes were corrected according to the rational nomenclature reported previously⁷.

7. From a literature citation and narrative framing standpoint, there are multiple categories of previous studies that must be cited and discussed, with regards to Cascade use for transcriptional control in eukaryotes; CRISPR-Cas system biology and crRNA processing, Type I systems (notably Pae Type I-F), and possibly more. Indeed, besides the aforementioned shortcomings regarding the published studies illustrating the potential of Type I systems for transcriptional control, the authors must provide more background and details and discussion on Type I CRISPR-Cas systems in general and Cascade and crRNA processing in particular. After all, it was first shown over a decade ago that CRISPR is driven by crRNA biogenesis by a Type I paper conspicuously missing from the cited references. Besides Type I-E systems, many early and also recent studies characterizing Type I-F CRISPR-Cas systems are missing, and it seems that PaeCascade has been more studied than the TfuCascade, as opposed to what the authors claim. This also applies to the section investing the impact of mismatch at periodic intervals along the crRN:targetDNA duplex.

Answer: We apologized for not making an intact and comprehensive citation and thank the reviewer for kindly pointing out our mistake. As, suggested, we have added more detailed information and cited references in the paragraph 2-4 of the introduction.

8. From a narrative standpoint, there are grammatical issues throughout the manuscript that warrant extensive narrative editorial corrections and improvements (see abstract for examples of missing words and grammatical shortcomings).

Answer: We have revised the article as suggested.

Reviewer #2:

Chen et al. report the development of a programmable CRISPRa system for gene activation using a Type I-F CRISPR-Cas system. Most reported implementations of CRISPRa are derived from class 2 systems. Type I systems present challenges because the RNA-protein complex that binds DNA includes multiple protein components, although there are potential advantages for recruiting multiple effectors via distinct subunits. The authors fused the VPR activator to various subunits of the Type I-F Cascade complex and demonstrate transcriptional activation at a heterologous reporter gene and multiple endogenous genes. They assess the effects of varying crRNA position, multiplexing crRNAs, varying crRNA spacer length, and crRNA-DNA mismatches.

A comparable Type I system was recently described in Nature Biotechnology (Pickar-Oliver et al., 2019 doi: 10.1038/s41587-019-0235-7; cited by the authors as ref 28). This paper demonstrated both transcriptional activation and repression, and included extensive genome-wide experiments to assess the specificity of the system. For transcriptional activation, this work used the p300 activator, although they also showed a comparison with the VPR activator. There is significant value in publishing multiple implementations of new CRISPRa systems; several Type II CRISPRa systems were published in prominent journals in 2013-2015.

The manuscript under review reports a set of unique and distinct findings that will be a valuable contribution to the literature. Most importantly they demonstrate that a Csy3-VPR fusion is an effective activator; Pickar-Oliver et al. did not report testing the analogous Cas7(Csy3)-p300 fusion. This finding is significant because multiple copies of Csy3 bind the crRNA spacer. By increasing the spacer length, the authors were able to increase gene activation, presumably by recruiting more copies of Csy3-VPR.

I have several comments that the authors should consider.

Major comments:

1. The authors should comment on the specificity of their system. While I do not think it is absolutely necessary to perform genome-wide sequence experiments (as in Pickar-Oliver et al.), the authors should include some gene activation experiments with off-target crRNAs. They should also briefly discuss the available literature on the specificity of Type I CRISPR-Cas systems.

Answer: We thank the reviewer for the thoughtful suggestion. As the new data above, we have performed the experiments as suggested. We designed three pairs of overlapping guides for SpCas9 and PaeCascade targeting *HBB* (two target sites) or *HBG* (one target site). We then picked all the possible off-target genes (≤ 4 base mismatches to either Cas9 gRNA or PaeCascade crRNA) and performed qPCR to test the specificity. As shown in **new Fig. 7** (Please see Fig. 2 above), we found *HBB* or *HBG* can be upregulated efficiently. At the same time, none of the putative off-target genes is activated, indicating that the specificity of dCas9-VPR and PaeCascade-VPR were comparable.

As suggested, we have also added a paragraph to discuss the specificity of other Type I systems.

2. On pg 8, the authors describe a two-plasmid delivery strategy for the Cascade complex proteins. The logic here is confusing on multiple levels. The motivation is that Csy3 is present in multiple copies and so might be limiting if delivered in the same plasmid as the other components. Based on this logic, it seems like the authors should deliver Csy3 on one plasmid and the other three proteins on a second plasmid, and then increase the ratio of Csy3 plasmid to the other plasmid. Instead, the authors delivered the proteins on two plasmids in pairs (Csy1/2 and Csy3/4), and they did not describe whether they varied the ratios of these two plasmids for transfections. Given the system described, it is unclear why there was any benefit to the helper strategy versus the 4-component single plasmid strategy. There is no explanation for how the helper plasmid strategy enabled the authors to change the ratio of Csy3 to the other components. Some additional experiments to address this point would be useful.

Answer: We thank the reviewer for the insightful suggestions. We have performed additional experiments according to the suggestions. We put the Csy1, Csy2, and Csy4 linked by P2A in one plasmid and the Csy3-VPR in another plasmid. These two plasmids (Csy1-Csy2-Csy4 and Csy3-VPR) were co-transfected into HEK293T with crRNA expressing plasmid targeting *HBB* or *HBG*. Different molar ratios of the Csy1-Csy2-Csy4 and Csy3-VPR

expressing plasmids were prepared and transfected as in **new Supplementary Figure 7a** (Please see Fig. 3a below). The activation of both two genes showed the highest efficiency at the most optimized molar ratio (1:1.5~1:6). Increasing the expression of Csy3 may be beneficial to transcription activation. However, it lowered the efficiency when other components were insufficient. We then compared the Csy1-Csy2-Csy4 and Csy3-VPR system (molar ratio = 1:3) with the 2-vector system (Csy1+Csy2 and Csy3-VPR+Csy4, molar ratio = 1:1) in which each Csy subunit was expressed separately with an independent promoter. *HBB* relative mRNA level of the 2-vector system was comparable to that of the Csy1-Csy2-Csy4 and Csy3-VPR system, while *HBG* relative mRNA level of the 2-vector system was three-fold higher than that of the Csy1-Csy2-Csy4 and Csy3-VPR system. Further experiments indicated that expressing the proteins on two plasmids driven by separated promoters (Csy1+Csy2/Csy3-VPR+Csy4) may be better, as shown in **new Supplementary Figure 7b** (Please see Fig. 3b below).

a

Ratio of Csy1-Csy2-Csy4 to Csy3-VPR

Ratio of Csy1-Csy2-Csy4 to Csy3-VPR

b

Fig.3. Activation of *HBB* or *HBG* with vectors encoding different arrangements of Csy subunits

(a) Csy1, Csy2 and Csy4 were expressed by P2A fusion in one plasmid and Csy3 in another. The two plasmids were transfected with different molar ratios.

(b) Comparison of the activation efficiency of two different constructs in activating *HBB* and *HBG*. Construct 1: Csy1, Csy2 and Csy4 were expressed by P2A fusion in one plasmid and Csy3 in another (molar ratio = 1:3). Construct 2: Csy1, Csy2 and Csy3-VPR, Csy4 were expressed in pairs on two plasmids (molar ratio = 1:1) with independent promoters. Data represented three biological repeats and displayed as mean \pm S.E.M. Statistical significance was calculated using one-way ANOVA (*, $P < 0.05$; **, $P < 0.01$; ***, $P < 0.001$).

3. At multiple points in the manuscript, the authors suggest that an advantage of the Type I system is that multiple subunits will facilitate novel applications by allowing multiple different tags to be recruited to the same complex. There are no experiments that justify this claim, although I agree that it is possible. In fairness, Pickar-Oliver et al. made the same claim, also without experimental justification. The authors could consider specifying exactly what types of multiple effector recruitment schemes might be desirable.

Answer: Thanks for your helpful suggestion. We have added the discussion "It is possible that we can activate gene expression with different kinds of effectors: Cascade-TET1 (Ten-Eleven Translocation dioxygenase1) fusion for DNA demethylation; Cascade-p300 fusion for histone acetylation; Cascade-VP64 or Cascade-VPR (VP64-p65-Rta) fusion for transcriptional factor recruitment, and achieve stronger and more persisted gene activation through combining these three methods properly(Chavez et al., 2016; Hilton et al., 2015; Liu et al., 2016; Morita et al., 2018)."

4. On pg 9, the authors compare their system to dCas9-VPR and suggest that their system is generally more effective. CRISPRa output can vary with sgRNA sequence and position relative to the TSS. It seems difficult to make a blanket conclusion about dCas9-VPR vs. the Type I-F system with the data shown. How did the authors choose the target sites for dCas9-VPR? Do they know if they are comparing the most effective sites between each system?

Answer: We agreed that CRISPRa output could vary with sgRNA sequence and position relative to the TSS. So for a fair comparison of the CRISPRa systems, we designed gRNAs and crRNAs for Cas9 and type I-F targeting the same, at least partial overlapping, regions in all the comparative experiments. And a more detailed comparison to dCas9-VPR, dAsCas12a, and EcoCascade-VPR was performed with overlapping gRNAs and crRNAs as shown in **new Fig. 3** (Please see Fig. 1 above). Our conclusion is that PaeCascade-VPR outperformed dCas9-VPR at least at some target sites.

Minor comments:

1. The introduction could be much more concise. There is more background description of class 1 and class 2 CRISPR systems than is really needed. Conversely, some details that might be helpful are missing. For example, it would be useful to explain that the Type I Cascade complex recruits a Cas3 nuclease, and that the system described here does not include the nuclease, just the DNA binding components of the Cascade complex.

Answer: We thank the suggestion of the reviewer, and we have revised the introduction as suggested. We have added, "the Class 1 type I CRISPR-Cas system relies on Cascade (CRISPR-associated complex for antiviral defense complex) for DNA binding, which further recruits Cas3 to degrade the foreign DNA."

2. In the last sentence of the abstract and in the last paragraph of the discussion, the authors suggest that this CRISPRa system could be useful in therapeutic applications. There are significant challenges for the use of any CRISPRa system as a therapeutic, including immunogenicity, delivery, and off-target effects. The authors do not indicate whether the Type I system described here might have any unique advantages compared to other CRISPRa systems. It is unclear why the authors are making this claim and whether it is justified by the specific findings reported in this paper.

Answer: We thank the reviewer for the kind suggestion, and we have revised the discussion as suggested. In the additional experiments, we found no activation of any predicted off-target genes by PaeCascade-VPR, as shown in new **Fig. 3** (Please see Fig. 1 above), which may provide some off-target insight into PaeCascade.

3. On pg 3, the authors write: "Comparing to current widely used class 2 systems, the multiple-subunit type I CRISPR has distinct property in genome editing, and the multiple subunit arrangement in the cascade raises the possibility of refining type I CRISPR into a "Swiss Army Knife" like platform for gene engineering." At this point in the manuscript, this statement is vague and confusing. The authors do a better job explaining their point on pg 4.

Answer: We apologize for the confusing statement. A distinct feature of Cascade-Cas3 CRISPR system for genome editing is the production of large fragment DNA deletion in the target site, which is an advantage of type I system for gene editing. As our research mainly focuses on developing the type I-F system for gene activation, the multiple Cas proteins in the Cascade provides a platform for different Cas protein-effector fusion strategies, in which different effectors, e.g. VPR or p300, can be fused to different Cas proteins to regulate gene expression. As your suggestion, we refined these descriptions in the newly revised manuscript to make the introduction much more concise understandable.

4. On pg 4, the authors write: "In contrast to Type I-E and I-B, *Pseudomonas aeruginosa* type I-F and *Shewanella putrefaciens* type I-Fv systems have fewer subunits and also bind dsDNA in bacteria." This statement is confusing. It reads as if the authors are saying I-E and I-B systems do not bind dsDNA, which is unlikely to be the point they are trying to make.

Answer: We apologized for our not appropriate description leading to the confusion and thank the reviewer for the suggestion. We have revised the article as suggested. We rewrote the statement into "In contrast to Type I-E and I-B, *Pseudomonas aeruginosa* type I-F and *Shewanella putrefaciens* type I-Fv systems require fewer subunits for dsDNA targeting in bacteria."

6. On pg 5, the authors write: "Comparing to current single effector class 2 systems, the multiple subunits in type I-F and type I-Fv might provide more combinations for tagging and increase signal strength if genetic modulators or tags are fused to different subunits. These characteristics and potential advantages make them better candidates as novel gene editing and regulating tools." The first sentence is plausible but the second sentence is speculation. There are many factors that could be considered to assess whether a system is "better". Also, note that here the authors are implying that having more subunits is better, but their justification for using I-F instead of I-E was that I-F has fewer subunits (pg. 4).

Answer: We apologized for our not appropriate description. Compared with class 2 systems, type I-E, type I-F and type I-Fv system have the advantages of multi-effector fusion and effector signal enhancement. And these Cascades are made up of 5, 4 and 3 subunits respectively. Thus, by comparison of these three systems, type I-F and type I-Fv Cascade will be easier to be manipulated when used for expression and genes activation in human cells. We have now changed the description into "Also, the multiple subunits in type I-F and type I-Fv might provide more combinations for tagging and increase signal strength when genetic modulators are fused to different subunits."

7. In Figure 1a, there are two arrows pointing to bands on the gel. The lower arrow is not described in the figure legend and does not seem to point to anything of significance.

Answer: We apologized for the improper marking and thanked the reviewer for kindly pointing out. We have removed the lower arrow.

8. In Figure 1b, where is the unshifted ssDNA on the gel?

Answer: The DNA on the gel was stained with GelRed. GelRed inserts and binds onto the DNA helix, therefore it prefers dsDNA more than ssDNA. dsDNA will be brighter than ssDNA at the same molar number and at the same UV exposure time. We have added "*" to indicate the unshifted ssDNA.

9. In Figure 1b, why is one of the samples here non-target ssDNA? In panel a, the corresponding sample is non-target dsDNA.

Answer: Thanks for the suggestion. We have now used a non-specific dsDNA control for PaeCascade (please see **new Fig. 1b**).

10. On page 9, the authors attempt to activate multiple endogenous genes. CRISPRa output can vary depending on the basal expression level of the target gene (e.g. doi:10.1038/nmeth.3871 & doi:10.1038/nature14136). Are similar trends occurring here? Can the authors offer any interpretation for why different outputs were observed with different endogenous gene targets?

Answer: Yes, similar trends occurred here. As suggested, we quantified the basal expression level of six genes (*HBB*, *HBG*, *IL1B*, *IL1R2*, *SOX2*, and *OCT4*) (new Fig 2d, please see Fig. 4 below). The activation fold change is correlated with the basal expression state of the genes, in which genes at low expression levels tend to have greater mRNA fold change in the activation assay. In addition, different outputs might also be caused by varied chromatin accessibility of each target. As higher-order chromatin environment and chromatin accessibility may influence the editing efficiency of Cas9^{8, 9}, the chromatin structure of certain gene in certain cell line might also play a role in CRISPR-based transcription activation systems.

Fig.4. Efficiency of target gene activation as a function of basal transcript levels.

Data from Fig. 2b was plotted by fold changes comparing to negative control and relative basal transcript level of *HBB*, *HBG*, *SOX2*, *OCT4*, *IL1B*, and *IL1R2*. Each dot represented the mean relative activation level of each crRNA.

11. On pg 11, the authors write: "The ability of PaeCascade-VPR to activate multiplex genes simultaneously with a single crRNA array instead of using individual crRNAs enormously simplifies the activation system, increasing the transfection efficiency and removing the roadblock of multiplex activation using type II CRISPR system." This statement is far too broad. There are many multiplex sgRNA delivery strategies that have been reported for Class 2 Type II CRISPR systems.

Answer: We thank the reviewer for your suggestion. We have revised the description in the manuscript to "makes it not necessary to express and deliver multiplex gRNAs independently in comparison with type II CRISPR system."

12. On pg 12, the authors describe how mismatches every 6th position are tolerated. This finding is interesting. Can the authors discuss further and provide any interpretation? Is there any precedent or structural rationale for this finding?

Answer: The structure of type I-F PaeCascade binding to dsDNA has been displayed and analyzed before¹⁰. The binding of the target strand to crRNA follows a periodic "5+1" pattern. The five consecutive base pairs followed by a one base pair gap which the two unpaired nucleotides between crRNA and target strand kink out in opposite directions. Therefore, the mismatch of per 6th nucleotide has less impact on target DNA binding and activation efficiency. The result of our experiment was consistent with its structure character. Thanks the reviewer for asking the question and we have revised the manuscript accordingly.

13. On pg 14, the authors write: "Furthermore, type I-B Cas7 (Csy3 equivalent) failed to induce gene activation when fused to transcription activator (28)." As best I can tell, ref 28 (Pickar-Oliver et al.) did not report testing a Cas7-p300 fusion. Can the authors clarify?

Answer: Thanks for pointing out. In that paper, the author failed to activate *ILIRN* with Cas7-p300 fusion LmoCascade in figure 4e (Pickar-Oliver et al.).

References

1. Morisaka H, *et al.* CRISPR-Cas3 induces broad and unidirectional genome editing in human cells. *Nature communications* **10**, (2019).
2. Cameron P, *et al.* Harnessing type I CRISPR-Cas systems for genome engineering in human cells. *Nature biotechnology* **37**, 1471-1477 (2019).
3. Pickar-Oliver A, *et al.* Targeted transcriptional modulation with type I CRISPR-Cas systems in human cells. *Nature biotechnology* **37**, 1493-1501 (2019).
4. Young JK, *et al.* The repurposing of type I-E CRISPR-Cascade for gene activation in plants. *Communications biology* **2**, 383 (2019).
5. Dolan AE, *et al.* Introducing a Spectrum of Long-Range Genomic Deletions in Human Embryonic Stem Cells Using Type I CRISPR-Cas. *Mol Cell* **74**, 936-950 e935 (2019).
6. Zetsche B, *et al.* Multiplex gene editing by CRISPR-Cpf1 using a single crRNA array. *Nature biotechnology*, (2016).
7. Makarova KS, Wolf YI, Koonin EV. Classification and Nomenclature of CRISPR-Cas Systems: Where from Here? *The CRISPR journal* **1**, 325-336 (2018).

8. Chen X, Liu J, Janssen JM, Goncalves M. The Chromatin Structure Differentially Impacts High-Specificity CRISPR-Cas9 Nuclease Strategies. *Molecular therapy Nucleic acids* **8**, 558-563 (2017).
9. Liu G, Yin K, Zhang Q, Gao C, Qiu JL. Modulating chromatin accessibility by transactivation and targeting proximal dsRNAs enhances Cas9 editing efficiency in vivo. *Genome biology* **20**, 145 (2019).
10. Guo TW, *et al.* Cryo-EM Structures Reveal Mechanism and Inhibition of DNA Targeting by a CRISPR-Cas Surveillance Complex. *Cell* **171**, 414-426 e412 (2017).

Reviewers' Comments:

Reviewer #1:

Remarks to the Author:

Chen et al. present a revised study of transcriptional control in human cells using a Type I-F CRISPR-Cas systems. They do show that Type I-F PaeCascade can be fused with a transcriptional activator and complexed with a targeting crRNA to increase transcriptional levels in HEK 293 cells, with higher efficiency than a few incumbent CRISPR-based alternatives.

As initially mentioned, there is limited novelty in light of multiple published reports on the use of Type I CRISPR-Cas systems for various outcomes in human and plant cells. The results are overall interesting, and the subtype I-F unique attributes noted, but have required extensive engineering and have relatively limited effect, and novelty. Added data regarding comparative analyses vs. other CRISPR effectors and specificity are noted and add value to the manuscript.

The added data for comparative analysis of various CRISPR effectors is noted and a noteworthy addition to the manuscript, though it is unclear how impactful the gains are compared to other platforms (at least in a few cases), and it is clear that there is variability across genes/targets/sequences, especially when taking into account the relative impact of guide selection on transcription levels (the select of overlapping guides is great and warranted, and does likely limit options for more comprehensive comparative analyses). The conclusions made by the authors are justified in light of the data presented, but I am still unsure how quantitatively and practically impactful these results will be for the community (both readership and users).

Improvements regarding addressing raised issues, expanding data and cited literature, clarifying statements and editing text are noted, and altogether improved the manuscript narrative. The extensive edits in the introduction section are noted. The latest CRISPR nomenclature by Makarova et al., should be cited.

Reviewer #2:

Remarks to the Author:

The authors have satisfactorily addressed the reviewer's comments.

The revised introduction is much improved. However, there is still probably more background than is needed. For example, pg 4 line 63 is irrelevant to the work described in this manuscript: "Mature crRNAs processed by *Escherichia coli* (*E. coli*) Cse3 and *Pyrococcus furiosus* (*P. furiosus*) Cas6 are terminated with 3'-end 2',3'-cyclic phosphate, while the crRNA grated by *Pseudomonas aeruginosa* (*P. aeru*) Csy4 has a terminal 3'-end phosphate."

I disagree with Reviewer #1 point 1. Reviewer #1 wrote: "These 4 manuscripts, and possibly more as well as a few additional studies presumably in press strongly negatively impact the novelty and impact of the narrative in general and results presented in particular." Reviewer #1 is correct that there is significant precedent in this area, but I think that there are important novel findings in this work that are impactful, as noted by the authors in their response. Further, I think it does not make sense to evaluate the novelty of a manuscript currently under review by referencing unknown manuscripts that might be in press.

REVIEWERS' COMMENTS:

Reviewer #1 (Remarks to the Author):

Chen et al. present a revised study of transcriptional control in human cells using a Type I-F CRISPR-Cas systems. They do show that Type I-F PaeCascade can be fused with a transcriptional activator and complexed with a targeting crRNA to increase transcriptional levels in HEK 293 cells, with higher efficiency than a few incumbent CRISPR-based alternatives.

As initially mentioned, there is limited novelty in light of multiple published reports on the use of Type I CRISPR-Cas systems for various outcomes in human and plant cells. The results are overall interesting, and the subtype I-F unique attributes noted, but have required extensive engineering and have relatively limited effect, and novelty. Added data regarding comparative analyses vs. other CRISPR effectors and specificity are noted and add value to the manuscript.

The added data for comparative analysis of various CRISPR effectors is noted and a noteworthy addition to the manuscript, though it is unclear how impactful the gains are compared to other platforms (at least in a few cases), and it is clear that there is variability across genes/targets/sequences, especially when taking into account the relative impact of guide selection on transcription levels (the select of overlapping guides is great and warranted, and does likely limit options for more comprehensive comparative analyses). The conclusions made by the authors are justified in light of the data presented, but I am still unsure how quantitatively and practically impactful these results will be for the community (both readership and users).

Improvements regarding addressing raised issues, expanding data and cited literature, clarifying statements and editing text are noted, and altogether improved the manuscript narrative. The extensive edits in the introduction section are noted. The latest CRISPR nomenclature by Makarova et al., should be cited.

Answer: We thank the reviewer for his/her interest in our work. We agree with the reviewer that the efficiency of Type I-F system may vary among different target sites, but this same problem is also true for other known CRISPR-Cas tools. Importantly, our study indicates that tagging the multi-copy subunit Csy3 enables stronger gene activation and thus out-performs the common CRISPR-Cas tools. We think our work is novel and impactful. As suggested, we have revised the manuscript and added the citation.

Reviewer #2 (Remarks to the Author):

The authors have satisfactorily addressed the reviewer's comments.

The revised introduction is much improved. However, there is still probably more background than is needed. For example, pg 4 line 63 is irrelevant to the work described in this manuscript: "Mature crRNAs processed by Escherichia coli (E. coli) Cse3 and Pyrococcus furiosus (P. furiosus) Cas6 are terminated with 3'-end 2',3'-cyclic phosphate, while the crRNA granted by Pseudomonas

aeruginosa (P. aeru) Csy4 has a terminal 3'-end phosphate.”

Answer: We thank the reviewer for the suggestion. We have revised the background and made the introduction more concise than the previous version.

I disagree with Reviewer #1 point 1. Reviewer #1 wrote: “These 4 manuscripts, and possibly more as well as a few additional studies presumably in press strongly negatively impact the novelty and impact of the narrative in general and results presented in particular.” Reviewer #1 is correct that there is significant precedent in this area, but I think that there are important novel findings in this work that are impactful, as noted by the authors in their response. Further, I think it does not make sense to evaluate the novelty of a manuscript currently under review by referencing unknown manuscripts that might be in press.

Answer: We agree and thank the reviewer for the insightful points.